# EGALITARIAN GRADIENT DESCENT: A SIMPLE APPROACH TO ACCELERATED GROKKING

**Ali Saheb Pasand**
McGill University & Mila Institute
ali.sahebpasand@mail.mcgill.ca

**Elvis Dohmatob**
Concordia University & Mila Institute
elvis.dohmatob@concordia.ca

## ABSTRACT

Grokking is the phenomenon whereby, unlike the training performance which peaks very early on during training, the test/generalization performance of a model stagnates over arbitrarily many epochs and then suddenly jumps to usually close to perfect levels. In practice, it is desirable to reduce the length of such plateaus, that is to make the learning process "grok" faster. In this work, we provide new insights into grokking. First, we show both empirically and theoretically that grokking can be induced by asymmetric speeds of (stochastic) gradient descent, along different principal (i.e singular directions) of the gradients. We then propose a simple modification that normalizes the gradients so that dynamics along all the principal directions evolves at exactly the same speed. Then, we establish that this modified method, which we call egalitarian gradient descent (EGD) and can be seen as a carefully modified form of natural gradient descent, groks much faster. In fact, in some cases the stagnation is completely removed. Finally, we empirically show that on classical arithmetic problems like modular addition and sparse parity problem which this stagnation has been widely observed and intensively studied, that our proposed method removes the plateaus[1].

## 1 INTRODUCTION

Neural networks sometimes exhibit a striking training dynamic known as *grokking*: after rapidly driving the training error to (near) zero, test performance can linger near chance for a long period before rising abruptly to near-perfect generalization—often without any change to the optimizer or explicit early stopping (Power et al., 2022). Grokking has been observed across architectures and tasks, from algorithmic problems such as modular arithmetic (Gromov, 2023; Doshi et al., 2024) and formal languages to more naturalistic settings (Liu et al., 2023; Nanda et al., 2023). Yet despite a rapidly growing body of empirical and mechanistic case studies, we still lack a principled account of *why* the delay occurs, *what* structural features crystallize at the transition, and *how* to predict or control it (see Section 2 for a detailed review of the existing literature).

In this work, we examine grokking through the lens of the dynamics of eigen-spectra of gradients during optimization, and propose a simple modification of (stochastic) gradient descent which provably reduces the length of the plateau without compromising the level of generalization of the model at the end of training.

**Contributions.** Our main contributions can be summarized as follows.

- *Egalitarian Gradient Descent (EGD).* We propose a novel and simple method for accelerated grokking and study its properties both theoretically and empirically. Our method operates by modifying the gradients at each layer so that the principal directions (aka singular directions) are conserved, but the speed of evolution of the dynamics along each of these directions is the same. This stabilizes the training by reducing the effect of ill-conditioned loss landscapes (where gradients vary significantly in magnitude across different principal directions), leading to accelerated grokking.
- *An Effective Theory.* We develop a theory which shows that our proposed EGD method is guaranteed to drastically grok, compared to vanilla methods such as (stochastic) gradient

---

[1]Code available at https://github.com/asahebpa/Egalitarian-Gradient-Descent

descent. We also make formal links to natural gradient descent. In fact, we show that our proposed method (which can be seen as a carefully modified version of natural gradient descent) is a simplified version of Grokfast (which operates by low-pass filtering of the gradients to boost weaker components). At a high-level, we also show that both methods have the same inductive bias of making the scales of the dynamics of the evolution of the parameters comparable along different important directions.

– *Empirical Verification.* We run extensive experiments on toy problems (binary classification of anisotropic multivariate data) and arithmetic problems (sparse parity, modular addition, etc.) and show that our proposed method typically groks immediately at the very beginning of the learning curve. To illustrate the computational efficiency, robustness, and generality of our EGD compared to benchmarks like Grokfast (Lee et al., 2024), we also provide a vast array of experiments in the appendix on more realistic datasets (MNIST, CIFAR-10), and models (MLPs, CNNs, transformers).

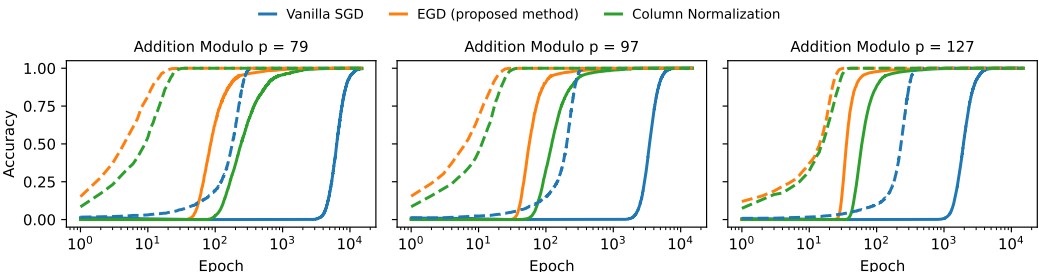

Figure 1: **Results on Modular Addition** for different values of the modulus $p$. Solid lines correspond to test accuracy and broken lines correspond to train accuracy. In all cases, our proposed EGD (*egalitarian gradient descent*) method groks after only a few epochs, while vanilla (stochastic) gradient descent stagnates for a long period before eventually grokking. We also include "Column Normalization", a simplification of EGD which simply rescales the columns of gradient matrices by dividing by their $L_2$ norm. Even this simplification seems to grok much faster than the baseline, vanilla (S)GD. Refer to Section 5 for details and to Appendix B for the hyper-parameters used.

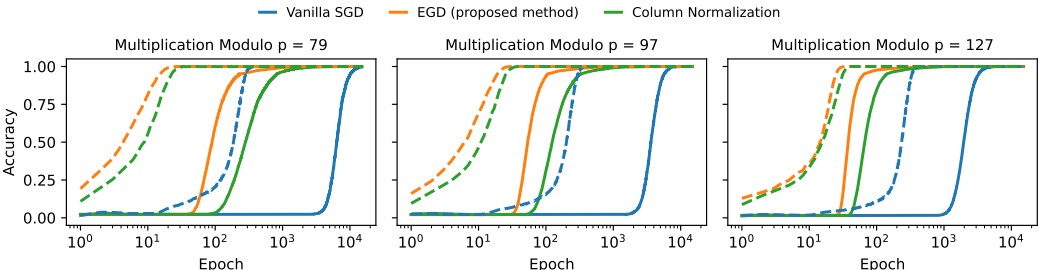

Figure 2: **Results on Modular Multiplication** for different values of the modulus $p$. Solid lines correspond to test accuracy and broken lines correspond to train accuracy. In all cases, our proposed EGD method groks after only a few epochs, while all the other methods stagnate a long period before eventually grokking. Refer to Section 5 for details and to Appendix B for the hyperparameters used.

## 2 RELATED WORK

Grokking—the phenomenon where models first overfit the training set and only much later undergo a sharp jump in test accuracy—was first documented by Power et al. (2022). Subsequent empirical studies broadened the scope beyond purely algorithmic datasets and analyzed conditions under which grokking appears or disappears, e.g., via loss–norm tradeoffs, optimizer choice and dataset/regularization choices (Liu et al., 2023; Nanda et al., 2023; Notsawo et al., 2025; Thilak et al., 2022; Liu et al., 2022; Davies et al., 2023; Notsawo et al., 2023; Varma et al., 2023).

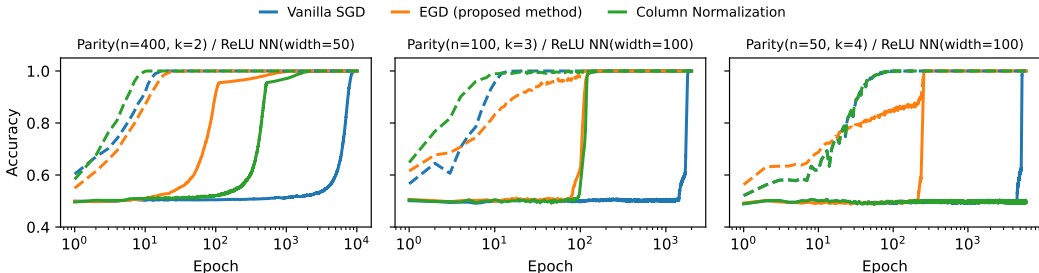

Figure 3: **Results on Sparse Parity Problem.** Solid lines correspond to test accuracy and broken lines correspond to train accuracy. All three plots show that our method (EGD) groks significantly faster than other methods. Refer to Section 5 for details on the experimental setup and to Appendix B for the hyperparameters used.

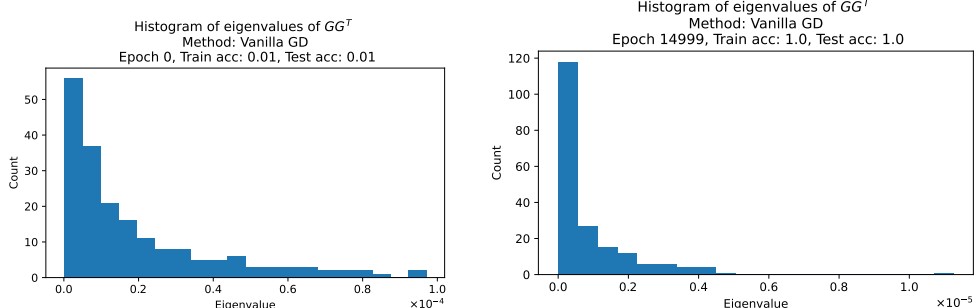

Figure 4: **Ill-conditioned Gradient Spectra** causes delayed generalization. We consider the problem of learning addition modulo 97 from data, with a two-layer ReLU neural network. At the start of optimization through to the end, the gradient matrix $G$ for the hidden layer has a poor condition number. Here, we see that the largest singular-value (corresponding to a fast direction) is much larger than the smallest (corresponding to slow directions). This causes the overall dynamics of vanilla (S)GD to stall for arbitrarily long times, leading to delayed generalization (see Figure 1). Our proposed method, EGD (*egalitarian gradient descent*) forces all the singular values of $G$ to be equal.

**Plateau Phenomena and Singular Learning Theory.** Long before grokking (Power et al., 2022), prolonged *training error* plateaus were analyzed in information geometry and statistical–mechanics treatments of neural nets. Plateaus arise near *singular* parameter regions—caused by permutation symmetries and redundancies—where the Fisher information degenerates and gradient flow becomes extremely slow (Wei et al., 2008; Amari et al., 2018). Classical online-learning studies of multilayer (soft-committee) networks likewise reported long transients and quantified their dependence on teacher–student alignment (Saad & Solla, 1995). Singular learning theory gives a general account via algebraic geometry, relating generalization behavior and marginal likelihood to model singularities (Watanabe, 2009). More recently, statistical–mechanical analyses have shown how input-data spectra modulate whether plateaus appear prominently at all (Yoshida & Okada, 2019). While these plateaus need not coincide with delayed generalization as in grokking, the underlying mechanisms—degenerate Fisher spectra, symmetry breaking, and slow modes—are highly relevant to grokking.

**Grokking in Modular Arithmetic.** A rich line of mechanistic interpretability work reverse-engineers the circuits by which small transformers and MLPs implement modular addition. Notably, Zhong et al. (2023) identify complementary algorithmic mechanisms ("clock" and "pizza") and circular embeddings that emerge at the grokking transition. Complementary analyses and constructive solutions for modular addition are provided in Gromov (2023), while Doshi et al. (2024) extend these insights to modular polynomials (e.g., multi-term addition and related arithmetic), showing that trained networks converge toward these circuits near the onset of generalization.

**Grokking as Kernel Escape.** Another perspective sees grokking as a dynamical transition from a lazy, kernel-like regime to a rich, feature-learning regime. Kumar et al. (2024); Walker et al. (2025) formalize conditions under which such a transition yields delayed generalization without requiring

explicit regularization. For modular addition, Mohamadi et al. (2024) give a theoretical analysis explaining why early kernel-regime learning cannot generalize under symmetry constraints, while later training escapes the kernel and finds small-norm, generalizing solutions. Aligned with this view, Varma et al. (2023) study grokking through the lens of circuit efficiency. This study shows that networks quickly learn memorizing solutions which generalize poorly, and the circuits that generalize well will take longer to form. When both circuits are formed, the generalizing circuits will be dominant in generating outputs.

**Training Stability and the Optimization Lens.** An orthogonal view emphasizes numerical stability and optimization dynamics as bottlenecks that can stall generalization; Prieto et al. (2025) argue that operating near the edge of numerical stability can induce grokking-like delays and propose remedies that restore or accelerate test performance. Similar to this perspective, Thilak et al. (2022) identify a mechanism called "Slingshot" in which cyclic phase transitions in adaptive optimizers co-occur with grokking. The early spectral characteristics of learning curves have also been proposed as predictors of grokking, reducing the need for long training (Notsawo et al., 2023). In a more general study, Liu et al. (2022) develop an effective theory of representation learning, showing that generalization emerges in a specific zone for the weights of a network called the "Goldilocks Zone". Grokking occurs when the weights enter this region, a narrow phase between memorization and confusion. In broader perspectives, grokking has been linked to phenomena such as double descent and emergent abilities (Huang et al., 2024; Davies et al., 2023), suggesting that delayed generalization can arise from competition between memorization and generalization dynamics. Recent theoretical analyses have studied grokking in high-dimensional linear and logistic regression models under Gaussian assumptions, relating delayed generalization to spectral properties of the empirical covariance near interpolation thresholds (Beck et al., 2025; Levi et al., 2024). Taken together, these studies show that both data and representation dependent dynamics play roles in the emergence of grokking phenomena. As a result, grokking can be affected by architectural, optimization dynamics, and data-related interventions.

**Grokking beyond arithmetic and Delay Mitigation.** Grokking is not limited to algorithmic tasks. It has been observed in some computer vision and natural understanding tasks (Liu et al., 2023; Lee et al., 2024). Also, structural grokking has been shown to occur in language models, where models discover hierarchical sentence structures after extensive training (Murty et al., 2023; Zhu et al., 2024). As more practical tasks exhibit grokking behavior, an important and interesting research question is how we can reduce the delay between memorization and generalization. Practical interventions have been shown to be effective to shorten or remove the grokking delay (Lee et al., 2024; Lyle et al., 2025). *Grokfast* amplifies slow (low-frequency) gradient components via simple optimizer-side filters, consistently accelerating grokking across tasks and architectures (Lee et al., 2024). In Section 4.2, we discuss the algorithmic and conceptual benefits of our proposed method over this Grokfast.

Also, accelerating grokking has shown to be beneficial in a practical scenario in which data distributions shift during training. To address this, Lyle et al. (2025) propose effective learning rate scaling and re-warming as a method to trigger and accelerate feature-learning dynamics during training which can both accelerate grokking and address the issue of primacy bias in continual learning. These dynamics-aware methods complement representation-side interventions and regularization/norm-control levers observed to modulate the phenomenon (Liu et al., 2023; Nanda et al., 2023).

## 3 WARM-UP: MOTIVATION FROM A SIMPLIFIED SETUP

We start with a simple analytically solvable example where the structure of the data (relative strength of features, relative difficulty of training and test datasets) and optimization choices (learning rate, size of initialization) interact to provably produce grokking curves with arbitrarily long plateaus. The setting we consider here is directly motivated by the observations in Figure 4, where delayed generalization in vanilla (S)GD is caused by ill-conditioned gradient matrices.

**Data Distribution.** Fix some small $\varepsilon > 0$ and let $z = (z^{(1)}, z^{(2)})$ be centered Gaussian random vector with covariance matrix $\Sigma = \begin{pmatrix} 1 & 0 \\ 0 & \varepsilon \end{pmatrix}$. Let $x_{train} \in \mathbb{R}^2$ be a folded version of $z$ obtained by conditioning on the event $|z^\top e_1| = |z^{(1)}| \geq s$, where $e_1 = (1, 0)$ is the horizontal basis vector in $\mathbb{R}^2$. Let $x_{test}$ be a folded version of $z$ obtained by conditioning on the event $(z^\top v)z^{(1)} \leq 0$. Here,

$v \in \mathbb{R}^2$ is a fixed unit-vector which makes an angle $\theta \in (0, \pi/2)$ with $e_1$. Thus, the condition number of the feature covariance matrix is $1/\varepsilon \gg 1$. This covariance ill-conditioning, captures, albeit in a simplified and instrumental/controllable manner, what is going on in Figure 4.

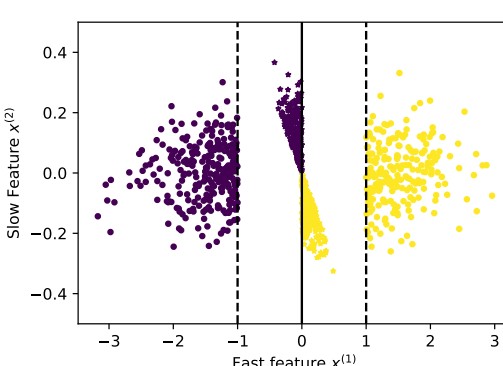

Figure 5: **A Toy Setup which Induces Stagnation in Gradient Descent (GD).** Training data points correspond to circles and test data points correspond to stars (middle region). The broken lines correspond to the large margin of the training data (their separation is $2s$), while the solid line is the ground-truth decision-boundary $x^{(1)} = 0$. The variance of the slow feature $x^{(2)}$ scales like $\varepsilon \ll 1$. GD would quickly find a linear model which perfectly separates the training data but will take a time of order $1/\varepsilon$ to find the ground-truth model which attains perfect test accuracy. See Figure 6.

The training data is $(x_1, y_1), \ldots, (x_n, y_n)$ where $y_i := \text{sign}(x_i^{(1)})$, and the feature vectors $x_i$'s are iid copies of $x_{train}$. Here, $x_i^{(j)} := x_i^\top e_j$ is the 1 component of the $i$th datapoint $x_i$. For the test data, the feature vectors are iid copies of $x_{test}$ instead. The larger the value of $s$, the easier it is to quickly memorize the training data (perfect training accuracy). The situation is illustrated in Figure 5

**Remark 1.** *In the above, there is a shift between the distribution of the training data and the test data. In Appendix A.3, we also consider the setting where the training distribution coincides with the test distribution, the distribution of $z$ constructed above.*

**Model.** We consider a linear model $x \mapsto \text{sign}(x^\top \hat{w})$, where the weights vector $\hat{w}$ is obtained by minimizing the quadratic loss function

$$\ell(w) := \frac{1}{n} \sum_{i=1}^n (x_i^\top w - y_i)^2 = \frac{1}{n} \|Xw - Y\|^2, \tag{1}$$

where $X \in \mathbb{R}^{n \times d}$ is the design matrix with rows $x_1, \ldots, x_n$, and $Y = (y_1, \ldots, y_n) \in \{\pm 1\}^n$ is response vector. We choose this loss function because it leads to tractable analysis while retaining the same phenomenology we would get using the logistic loss function, for example.

### 3.1 VANILLA GRADIENT-DESCENT DYNAMICS

With step size $\eta$, the vanilla gradient-descent (GD) on the loss (1) gives the following recursion

$$w(k) = w(k-1) - \eta X^\top (Xw(k-1) - Y)/n = Aw(k-1) + b, \tag{2}$$

$$\text{with } b := \eta X^\top Y/n, \quad A := I - \eta \hat{\Sigma}, \quad \hat{\Sigma} := X^\top X/n. \tag{3}$$

Here $\eta > 0$ is a sufficiently small stepsize / learning rate, $k$ is the iteration (aka *epoch*), and $\hat{\Sigma}$ is the empirical covariance matrix. We can explicitly solve the above recursion to get (see the appendix)

$$w(k) = A^k w(0) + (I - A^k)\hat{w}_{ols}, \tag{4}$$

where $\hat{w}_{ols} := X^+ Y = \hat{\Sigma}^{-1} X^\top Y/n$ is the ordinary least-squares solution. Thus, $w(k)$ interpolates between the initialization $w(0)$ and the least-squares solution $\hat{w}_{ols}$.

Define constants $m_1 = m_1(s) \in (0, 1)$, $m_2 = m_2(s) > 1$, $\alpha = \alpha(s) \in \mathbb{R}$, $\beta = \beta(\varepsilon) \in (0, 1)$ by

$$m_1 := \mathbb{E}[|x_i^\top e_1|] = \varphi(s)/Q(s), \ m_2 := \mathbb{E}[|x_i^\top e_1|^2] = 1 + sm_1, \ \alpha := 1 - \eta m_2, \ \beta := 1 - \eta \varepsilon, \tag{5}$$

where $\varphi$ is the PDF of the standard Gaussian distribution, and $Q := 1 - \Phi$ is its survival function. Let the initialization be $w(0) = u = (u_1, u_2)$, and define the following dynamical quantities

$$\mu_k := \alpha^k u_1 + (1 - \alpha^k)m_1/m_2, \ \nu_k := \beta^k u_2, \ L_k := \sqrt{\mu_k^2 + \varepsilon \nu_k^2}, \ r_k := \mu_k/\nu_k. \tag{6}$$

The evolution of the test error is given analytically by the following result.

**Theorem 1.** *For large $n$, it holds w.h.p that: for any iteration $k \geq 1$,*

$$E(w(k)) \simeq \min(1, \arccos(r_k)/\arccos(r)), \text{ with } r := \rho/\gamma, \quad r_k := \mu_k/L_k, \quad (7)$$

$$\text{where } \rho := \cos\theta, \quad \gamma := \sqrt{\rho^2 + \varepsilon \cdot (1 - \rho^2)}, \quad L_k := \sqrt{\mu_k^2 + \varepsilon\nu_k^2}. \quad (8)$$

Theorem 1 reveals that: the test error plateaus at value $100\%$, until $k$ is sufficiently large that $r_k$ decreases below $r$, at which point the error starts decreasing at the rate $\arccos(r_k)/\arccos(r)$.

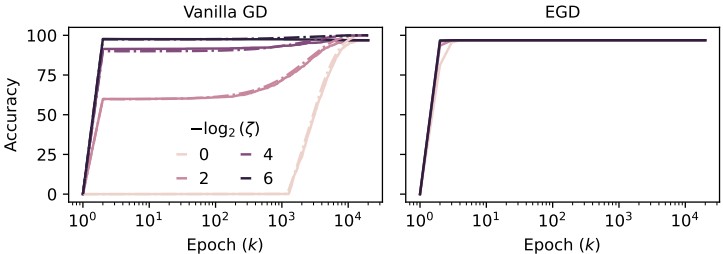

Figure 6: **Grokking on the Toy Problem.** Solid lines correspond to experimental results, while broken lines correspond to our theory (Theorem 1). The initialization is $w(0)$ is such that $\|w(0)\| = \zeta$. Thus, the scalar $\zeta > 0$ controls the size of the initialization. **Left.** As predicted by Theorem 1 and Corollary 1, large initialization leads to delayed generalization in vanilla GD, i.e. long plateaus (of length $k_* \asymp 1/\varepsilon$) of stagnation in the test performance, while small initialization attenuates it ($k_* \asymp \log 1/(\tau\varepsilon)$, where $\tau \propto \zeta$). Note that for this problem, the ratio $1/\varepsilon \gg 1$ represents the condition number of the covariance matrix of the features. **Right.** Our proposed EGD (*egalitarian gradient descent*) method groks immediately: the generalization error jumps to a perfect value after only a few iterations. Moreover, EGD appears to be completely insensitive to the scaling of the initialization, which is a desirable property in real-world optimization.

We have the following important corollary.

**Corollary 1** (The distribution-shift case). *Fix a constant $c > 1$. We have the following phase diagram: (A) **Large Initialization.** If $|u_2|$ is large in the sense that $\tau := |u_2| |\tan\theta| m_2/m_1 > c$, then the plateau length of the test error (i.e., the time to grok) is of order $k_* \asymp \frac{\log\tau}{\eta\varepsilon} \asymp \frac{1}{\varepsilon}$.*

*(B) **Small Initialization.** If $|u_2|$ is small, then the plateau length is of order $k_* \asymp \frac{1}{\eta} \log\frac{1}{\tau\varepsilon} \asymp \log\frac{1}{\varepsilon}$.*

Corollary 1 shows that a model trained via vanilla gradient descent will quickly converge to a decision-boundary which memorizes the training data ($100\%$ training accuracy, very poor test accuracy), but it will take a long time of order $1/\varepsilon$ or $\log 1/\varepsilon$ (depending on the size of initialization) before it converges to the true decision-boundary (Theorem 1), and only then does the test error abruptly increase to $100\%$. This result can be extended to vanilla stochastic gradient descent (SGD).

**The Case of Same Train and Test Distribution.** The setup (see Remark 1) is analyzed in Appendix A.3, and we get the same order of magnitude, $1/(\eta\varepsilon)$, for the length of the plateau in the case of vanilla GD.

**The Issue with Vanilla Gradient Descent.** One can show that for large sample size ($n \to \infty$), the empirical covariance matrix $\hat{\Sigma}$ which modulates the dynamics (2) has the following approximation.

$$\hat{\Sigma} \simeq \begin{pmatrix} m_2 & 0 \\ 0 & \varepsilon \end{pmatrix},$$

Where $m_2$ is as defined in (5). Since $m_2$ is a fixed positive constant, the condition number of the RHS is of order $1/\varepsilon$. This controls the rate at which the GD iterates $w(k)$ converge to the least-squares solution via (4), which is in fact optimal (perfect test accuracy) here since we are in the infinite sample regime). Therefore,

> **Insight #1.** *Grokking profile for this toy problem is therefore solely due to a delayed convergence least-squares solution caused by ill-conditioned gradients.*

## 3.2 Accelerated Grokking via a Modified Gradient Scheme

Consider the following modified dynamics, which will later form the basis for our proposed method:

$$w(k) = w(k-1) - \eta \hat{\Sigma}^{-1} X^\top (Xw(k-1) - Y)/n, \tag{9}$$

with $\hat{\Sigma} := X^\top X/n$ as before. Expanding the above equation, the dynamics become

$$w(k) = w(k-1) - \eta \hat{\Sigma}^{-1}((X^\top X/n)w(k-1)) - X^\top Y/n) = aw(k-1) + \eta \hat{w}_{ols},$$

with $a := 1 - \eta$. What has happened is that the inverse FIM has killed the ill-conditioned $\hat{\Sigma}$ matrix which was multiplicatively damping the iterates. Solving the above recurrence gives

$$w(k) = a^k w(0) + (\sum_{j=0}^{k-1} a^j)\eta \hat{w}_{ols} = a^k w(0) + \frac{1-a^k}{1-a}\eta \hat{w}_{ols} = a^k w(0) + (1-a^k)\hat{w}_{ols}. \tag{10}$$

The troublesome dependence on $\varepsilon$ has now completely disappeared from the picture. Moreover, we see that the convergence to $\hat{w}_{ols}$ is now much faster than with the vanilla dynamics (4), namely $w(k) = A^k w(0) + (I - A^k)\hat{w}_{ols}$, where $A := I - \eta \hat{\Sigma}$. Thus, in (10) an isotropic matrix $aI$ (spectral radius = $a = 1 - \eta$, a fixed positive constant less than 1 for any stepsize $\eta \in (0,1)$) replaces

$$A = I - \eta \hat{\Sigma} \simeq \begin{pmatrix} 1 - \eta m_2 & 0 \\ 0 & 1 - \eta \varepsilon \end{pmatrix}$$

which has spectral radius = $1 - \eta \epsilon \approx 1$, in (4). Therefore, under the modified gradient descent update rule (9), the iterates $w(k)$ converge to $\hat{w}_{ols}$ at an exponential rate which is independent of $\varepsilon$. This leads to grokking after just a few iterations, as seen in Figure 6.

> **Insight #2.** *The modified GD update rule* (9) *induces the desirable normalizing effect whereby the optimization dynamics has exactly the same speed along all principal directions.*

## 4 Proposed Method: Egalitarian Gradient Descent

Motivated by the insights from Section 3.2, we now consider a general non-linear model (neural network, etc.) on a general problem and let $G$ be the gradient matrix for an arbitrary layer. Thus, $G$ has shape $m \times p$, where $m$ is the fan-out and $p$ is the fan-in width for that layer. For example, if the layer is the hidden layer in a two-layer feedforward full-connected neural network, then $m$ is the number of hidden neurons and $p$ is the input-dimension. Consider the following transformation of the gradient matrix $G$:

$$\textbf{(EGD)} \qquad \tilde{G} := F^{-1/2}G = (GG^\top)^{-1/2}G, \tag{11}$$

where $F := GG^\top$ is the empirical Fisher matrix and $F^{-1/2}$ denotes its square root. We call this transformation *egalitarian gradient descent (EGD)*, a name that will become clear shortly. Observe that the above transformation leaves the left and right singular-vectors of $G$ unchanged but makes all the singular-values equal. Indeed, consider the singular-value decomposition (SVD) of $G$:

$$G = USV^\top = s_1 u_1 v_1^\top + s_2 u_2 v_2^\top + \dots, \tag{12}$$

where $U = (u_j)_j$ and $V = (v_j)_j$ contain the left and right singular-vectors (aka principal directions) of $G$, and $S = \text{diag}(s_1, s_2, \dots)$ contains its singular-values. Then, $(GG^\top)^{-1/2} = US^{-1}U^\top$ and so

$$\tilde{G} = (GG^\top)^{-1/2}G = US^{-1}U^\top USV^\top = US^{-1}SV^\top = UV^\top,$$

which has the same left and right singular-vectors as $G$ but singular-values all equal to 1. This means that *all through the optimization process, no principal direction will evolve faster/slower than another*. This justifies the name EGD (egalitarian gradient descent) given to (11), and we shall show that it drastically accelerates grokking.

**Remark 2.** *Note that in formula* (11)*, we have implicitly assumed that $G$ has full-rank. In case of rank deficiency (which will happen if $m > p$ for example), we simply replace $(GG^\top)^{-1}$ and $S^{-1}$ by their Moore-Penrose pseudo-inverses.*

**Practical Considerations.** The gradient $\tilde{G} = UV^\top$ for our proposed EGD method is obtained by performing SVD on the original gradient matrix $G$. This is the main computational cost incurred by our method. In practice, we turn off EGD and switch it for vanilla (S)GD once we detect grokking has occurred, by monitoring the validation loss. Moreover, additional experiments (refer to Appendices C and D) suggest that the SVD does not have to be precise, and approximate versions thereof, like randomized SVD (RSVD), work just fine, and lead to better stability and faster wall-clock time.

## 4.1 CONNECTION TO NATURAL GRADIENT DESCENT

The empirical *Fisher information matrix (FIM)* for any layer with gradient matrix $G$ is given by $F = GG^\top$, the same matrix which appears in (11). It is then easy to see that

$$\|\tilde{G}\|_F^2/m = (1/m)\operatorname{tr}[\tilde{G}^\top F\tilde{G}] = (1/m)\operatorname{tr}[(GG^\top)^{-1/2}G^\top GG^\top G(GG^\top)^{-1/2}] = (1/m)\operatorname{tr}I = 1.$$

Thus, the Fisher-norm of the modified gradient matrix $G$ is constant of motion of the dynamics induced by our proposed transformation (11). Note however that our EGD proposed method is not equivalent to *natural gradient descent (NGD)* (Amari, 1998; Pascanu & Bengio, 2013; Ollivier et al., 2017), which would correspond to $\bar{G} := F^{-1}G = (GG^\top)^{-1}G$. Notwithstanding, the two methods are linked like so

$$\underbrace{\tilde{G}}_{\text{EGD}} = \underbrace{F}_{\text{FIM}}^{1/2}\underbrace{\bar{G}}_{\text{NGD}}. \tag{13}$$

Therefore, our proposed EGD corresponds to a whitened version of NGD. The effect of this whitening is precisely to equalize the singular-values of the gradient matrix, leading to accelerated grokking.

## 4.2 COMPARISON WITH GRADIENT-FILTERING (GROKFAST)

The *Grokfast* method proposed by Lee et al. (2024) for inducing fast grokking goes as follows. Each row of the gradient matrix $G$ is replaced by $g + F(g)$, where $F(g)$ is a low-pass filtered version of $g$, computed by aggregating with a large buffer of the past history of gradients. This has the desirable effect of boosting the low-frequency components of the gradient, and attenuating the high-frequency components thereof.

Now, let $c_j := g^\top u_j$ be the $j$th component of $g$ measured in the eigen-basis for $G^\top G$. From (12),

$$(GG^\top)^{-1/2}g = (c_1/s_1)u_1 + (c_2/s_2)u_2 + \ldots \tag{14}$$

This down-weights the components of $g$ aligned with large $s_j$—the "high-frequency" directions—so our EGD update inherits the filtering inductive bias of Grokfast (Lee et al., 2024) as a by-product. Crucially, unlike Grokfast, EGD *equalizes* the optimization speed across principal directions, yielding isotropic progress in the eigenspace. EGD also enjoys the following important properties.

- *Memory.* In contrast to Grokfast (Lee et al., 2024), which maintains a large buffer of past gradients, our proposed EGD method incurs **no** additional memory overhead beyond the current gradient (and, if used, a running spectral estimate).

- *Simple and Hyperparameter-free.* EGD introduces no extra tuning knobs. It therefore avoids the task-dependent, time-consuming hyperparameter sweeps that are important for Grokfast. The formula for EGD (11) is a lightweight, drop-in modification of (stochastic) gradient descent with a closed-form, per-step rescaling in the principal basis; it requires no schedulers, no momentum variants, and no buffering.

- *Theoretical foundations.* EGD comes with a spectral analysis guaranteeing equalized per-mode convergence rates and, consequently, an accelerated exit from the test-error plateau (i.e., faster grokking). By comparison, Grokfast is a heuristic frequency filter without such guarantees.

### 4.3 RELATION TO MUON

At the level of update geometry, our proposed EGD method is related to the recently proposed Muon optimizer (Jordan et al., 2024): both replace an ill-conditioned gradient matrix with an approximately orthogonal/polar update, thereby homogenizing progress across singular directions. However, the two methods differ in both motivation and implementation. Muon applies Newton-Schulz iterations (Bernstein & Newhouse, 2024) to momentum updates and has been successfully adopted for training small language models (Jordan et al., 2024), with subsequent extensions to large-scale LLM pretraining (Liu et al., 2025; Team et al., 2025). EGD, instead, is derived from a spectral theory of grokking, where the preconditioning structure emerges as a principled theoretical consequence rather than an empirical design choice. From an implementation perspective, EGD applies SVD- or randomized-SVD-based normalization directly to gradients. As demonstrated in this paper, the proposed method is effective in accelerating grokking and also in improving adaptability in non-stationary settings when used in conjunction with a wide range of optimizers, both with and without momentum (see Sections 5, C, and E).

The implementation differences mentioned above lead to several practical distinctions between the two approaches that are worth discussing. When gradients exhibit low-rank structure, randomized SVD can exploit that by working with a truncation rank smaller than the matrix rank, reducing computational cost relative to full SVD while simultaneously providing explicit access to the singular value spectrum and allowing the truncation rank to vary across layers or stages of training. Newton-Schulz orthogonalization, by contrast, neither exposes singular values nor explicitly leverages low-rank structure, and instead relies on repeated large matrix multiplications at each iteration. Moreover, instability has been observed for Newton-Schulz-based updates in certain pretraining regimes (Team et al., 2025). Investigating whether randomized SVD exhibits similar instability would therefore be an interesting direction for future work.

The fact that two independently motivated methods converge to a similar geometric operation may itself be viewed as evidence for the effectiveness of gradient spectral normalization as an optimization primitive. This connection also suggests several promising directions for future work. Although Muon has shown strong empirical performance in LLM pretraining, its effectiveness in RL post-training algorithms has not yet been explored. Our results on non-stationarity and adaptability, particularly when EGD is combined with Adam (the dominant optimizer in RL post-training), suggest that gradient spectral normalization may improve sample efficiency in such settings (see Appendix E). Furthermore, because randomized SVD provides direct access to the singular value spectrum, an interesting direction would be to investigate its use as an alternative to Newton-Schulz iterations in LLM pretraining, especially in scenarios where gradients are effectively low-rank and the effective rank may vary across models, layers, or stages of training (Zhao et al., 2024).

## 5 EXPERIMENTAL VALIDATION

### 5.1 SPARSE PARITY PROBLEM

This is a well-known hard problem in statistical learning theory (Barak et al., 2022). Also, recent works have shown that this problem induces grokking in (stochastic) gradient descent (Merrill et al., 2023). An instance $\text{Parity}(n, k)$ of this problem is as follows. $n$ and $k$ are positive integers with $k \leq n$. A random $k$-element subset $S$ of $[n]$ is drawn once and for all, and then $N$ iid samples $(x_1, y_1), \ldots, (x_N, y_N)$ are generated, where the $x_i$ are iid uniform $n$-bit strings, and each $y_i$ corresponds to the XOR of the bits of $x_i$ restricted to the secret subset $S$, i.e $y_i := (-1)^{\sum_{j \in S} x_{ij}}$. The accuracy of a model $f : \{0, 1\}^n \to \{-1, 1\}$ is counting the proportion of points in a large held out test dataset (generated from the same distribution) have the true labels correctly predicted.

For the model, we consider a two-layer ReLU network $f(x) = \text{sign}(v^\top \sigma(W x))$ trained by optimizing hinge loss and weight decay with using different optimization strategies, on the parity problem with different values of $n$ and $k$: $(n, k) \in \{(400, 2), (100, 3), (50, 4)\}$. The batch size is set to 32. For reproducibility, information about low-level details like learning rate, amount of weight decay, etc. is provided in Appendix B. We compare different optimization strategies: vanilla (stochastic) GD, our proposed method EGD (11) (applied only on the gradient matrix $G$ for the hidden layer weights $W$), and an even simplified version of EGD where we replace each column of $G$.

The results are shown in Figure 3. As predicted by our theory, EGD groks very early on in the optimization process (typically after only a few epochs). In contrast, vanilla (S)GD goes through an arbitrarily long plateau of stagnation of the test error before eventually grokking.

## 5.2 MODULAR ARITHMETIC

Another family of problems that exhibits grokking behavior is modular arithmetic. This class of problems has extensively been studied in vast array of papers on grokking (Power et al., 2022; Liu et al., 2023; Lee et al., 2024; Mohamadi et al., 2024; Nanda et al., 2023; Notsawo et al., 2023; Zhong et al., 2023; Gromov, 2023; Doshi et al., 2024; Prieto et al., 2025). To formally define this class of problems, an instance $\text{Mod}(p, o)$ is defined by a prime modulus $p$ and an operation $o \in \{+, \times\}$ over $\mathbb{Z}_p$. Training data consists of $N$ i.i.d. samples $(x_i, y_i)$, which are a fraction of all $p^2$ possible combinations. Here $x_i = (a_i, b_i)$ is drawn uniformly from $\{0, 1, ..., p\} \times \{0, 1, ..., p\}$, and the label is calculated as $y_i = (a_i \ o \ b_i) \bmod \ p$. We train a two-layer ReLU network with cross-entropy loss and weight decay, using the same setup as in the sparse parity experiments. The complete set of hyperparameters is provided in Appendix B. Similar to sparse parity, we compare vanilla (stochastic) GD, our proposed method EGD (11), and a simplified column-wise variant of EGD.

As shown in Figures 1 and 2, EGD groks considerably earlier than other methods, achieving high accuracy after only a few epochs in both modular addition and multiplication tasks.

## 5.3 ADDITIONAL EXPERIMENTS: COMPUTATIONAL EFFICIENCY AND ROBUSTNESS

Appendices C, D, and E present results of additional experiments in which we test the usage of EGD across different model types (MLPs, CNNs, transformers) and more realistic datasets (MNIST and CIFAR-10). We have also compared EGD with Grokfast (Lee et al., 2024) as the strongest baseline. These experiments continue to show that our method leads to faster grokking while being computationally efficient. Experiments in Appendix E also show that EGD is robust in the presence of non-stationarity in the training data (encountered in reinforcement learning applications, for example).

## 6 CONCLUDING REMARKS

We studied grokking through the lens of gradient eigen-spectral dynamics and proposed *Equalitarian Gradient Descent (EGD)*, a simple, hyperparameter-free modification of stochastic gradient descent that equalizes optimization speed across principal directions which can be efficiently estimated via inexact SVD (e.g., randomized SVD). By down-weighting high-frequency components while preserving progress on slow, symmetry-aligned modes, EGD provides a principled, spectrum-aware update that consistently shortens the test-accuracy plateau without degrading final performance. Beyond empirical gains, our analysis clarifies how progress along low-frequency, task-aligned directions governs the timing of the generalization jump, and it yields compact diagnostics that relate early gradient spectra to later test improvement. The method is optimizer agnostic, easy to integrate into existing training loops, and introduces no additional tuning knobs.

Our proposed EGD method is lightweight and straightforward to deploy, offering faster grokking with minimal computation, and no additional memory usage unlike Grokfast (Lee et al., 2024).

**Future Work.** In future, we also plan to study plug and play combinations with adaptive optimizers and weight decay, behavior under nonstationary data and curriculum schedules, and broader benchmarks beyond algorithmic tasks. On the theoretical front, beyond the simplified setup considered in Section 3, it would be nice to have a convergence analysis of EGD for complex models like MLPs and transformers. This is highly difficult, but we have some preliminary path to such an analysis using ideas from spiked random matrix phase transitions (Baik et al., 2004).

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

# Appendix for "Egalitarian Gradient Descent: A Simple Approach to Accelerated Grokking"

CONTENTS

## A   PROOFS

### A.1   ANALYTIC THEORY FOR THE TOY PROBLEM

**Proof of Theorem 1.**   First observe that the vanilla gradient descent dynamics can be expanded as follows

$$
\begin{aligned}
w(k) &= w(k-1) - \eta X^\top (Xw(k-1) - Y)/n = Aw(k-1) + b \\
&= A^2 w(k-2) + Ab + b = \ldots = A^k w(0) + (A^{k-1} + \ldots + A + I)b \\
&= A^k w(0) + (I - A^k)(I - A)^{-1} b = A^k w(0) + (I - A^k)\eta^{-1}\hat{\Sigma}^{-1} b \\
&= A^k w(0) + (I - A^k)\hat{w}_{ols}.
\end{aligned}
$$

Now, for $z \sim \mathcal{N}(0, \Sigma)$ and $z_0 := \Sigma^{-1/2} z \sim \mathcal{N}(0, I_d)$, we can write

$$
\begin{aligned}
E_{test}(\hat{w}(k)) &= \mathbb{P}((z^\top e_1)(z^\top \hat{w}(k)) \le 0 \mid (z^\top e_1)(z^\top v) \le 0) \\
&= \mathbb{P}((z_0^\top \bar{e}_1)(z_0^\top \hat{\bar{w}}(k)) \le 0 \mid (z_0^\top \bar{e}_1)(z_0^\top \bar{v}) \le 0) \\
&= \frac{\arccos(r_{e_1,v}) + \arccos(r_{\hat{w}(k),e_1}) - \arccos(r_{\hat{w}(k),v})}{2\arccos(r_{e_1,v})} \\
&= \frac{1}{2}\left(1 - \frac{\arccos(r_{\hat{w}(k),v}) - \arccos(r_{\hat{w}(k),e_1})}{\arccos(r_{e_1,v})}\right),
\end{aligned}
$$

where $\bar{w} := \Sigma^{1/2} w$ and $r_{w,v} := \bar{w}^\top \bar{v}/(\|\bar{w}\|\|\bar{v}\|)$ is the cosine of the angle between $w$ and $v$, relative to the inner-product structure induced by $\Sigma$. The third line in the above display is a direct application of standard orthant probability formulae for bi-variate Gaussian random variables.

Note that $\bar{e}_1 = e_1$, and so $r_{e_1,v} = \cos(\theta)/\|v\| = \rho/\gamma =: r$. The result then follows upon invoking Lemma 2 to estimate $r_{\hat{w}(k),v}$ and $r_{\hat{w}(k),e_1}$.    □

**Proof of Corollary 1 (Grokking Profile of Vanilla GD).** We know that

$$\mu_k = \alpha^k u_1 + (1 - \alpha^k) m_1/m_2 = \alpha^k (u_1 - m_1/m_2) + m_1/m_2 \asymp m_1/m_2,$$

because the second term dominates. We get

$$r_k = \mu_k/L_k \asymp \frac{1}{\sqrt{1 + (\beta^k u_2 m_2/m_1)^2 \varepsilon}}.$$

For $u_1 = O(1)$, we have $\mu_k \geq 0$ for all $k$ and we can further write

$$r_k \simeq \frac{1}{\sqrt{1 + (u_2 \nu_k/\mu_k)^2 \varepsilon}}.$$

To ensure the test error is $E_{test}(w(k))$ is significantly better than chance level, we need $r_k < r$, which translates to

$$(\beta^{2k}/(1 - \alpha^k)^2)(u_2 m_2/m_1)^2 < 1/r^2 - 1 = \varepsilon \cdot (1 - \rho^2)/\rho^2 = \varepsilon |\tan \theta|^2,$$

or equivalently, in log space

$$k \geq k_* = \frac{\log \tau}{\log 1/\beta}, \text{ with } \tau := |u_2||\tan \theta| m_2/m_1.$$

Note that we have used the fact that $\log 1/\beta = -\log(1 - \eta \varepsilon) \asymp \eta \varepsilon$, for small $\varepsilon$. This gives

$$k_* \asymp \frac{1}{\eta \varepsilon} \log \tau$$

The case of small $|u_2|$ follows a similar argument to analyze the growth rate of $\arccos(r_k)$ and ultimately get $k_* \asymp (1/\eta) \log \frac{1}{\tau \varepsilon}$. $\qquad \square$

## A.2 IMPORTANT LEMMAS

The following lemmas are easily proved via the law of large numbers.

**Lemma 1.** *For large $n$, we have the deterministic approximations*

$$\hat{\Sigma} \simeq \begin{pmatrix} m_2 & 0 \\ 0 & \varepsilon \end{pmatrix}, \quad \frac{1}{n} X^\top Y \simeq \begin{pmatrix} m_1 \\ 0 \end{pmatrix}, \quad \hat{w}_{ols} \simeq \begin{pmatrix} m_1/m_2 \\ 0 \end{pmatrix}, \tag{15}$$

*where the notation "$\simeq$" ignores fluctuations of order $O_\mathbb{P}(n^{-1/2})$ in spectral or $L_2$ norm.*

The following lemma is a direct consequence of the previous via equation (4).

**Lemma 2.** *For any deterministic vector $v \in \mathbb{R}^2$, it holds that*

$$A^k \simeq \begin{pmatrix} \alpha^k & 0 \\ 0 & \beta^k \end{pmatrix}, \quad w(k) \simeq \begin{pmatrix} \mu_k \\ \nu_k \end{pmatrix}, \quad w(k)^\top \Sigma w(k) \simeq L_k^2, \quad w(k)^\top \Sigma v \simeq \mu_k v_1 + \varepsilon \nu_k v_2, \tag{16}$$

*where the notation "$\simeq$" ignores multiplicative factors of order $1 + O_\mathbb{P}(n^{-1/2})$.*

## A.3 THEORETICAL ANALYSIS OF TOY MODEL: THE EQUI-DISTRIBUTION CASE

We now consider a version of the toy setting considered in Section 3, with the following modification: the training and testing distributions are equal, consistently with Section 5. We shall establish Theorem 2, which is a counterpart of Corollary 1 in this equi-distributional setting.

Let $z \sim \mathcal{N}(0, S)$ with $S = \text{diag}(1, \varepsilon)$, $u = e_1$ (the first canonical basis vector of $\mathbb{R}^2$), $v = (\cos \theta, \sin \theta)$, $\theta \in [0, \pi]$. The distribution $P$ (of both training and testing data!) is the law of $(x, y)$ where $x$ has the same distribution as $z$ conditioned on the event $(z^\top u)(z^\top v) \leq 0$ and $y = \text{sign}(x^\top u)$.

The situation is illustrated in Figure 7.

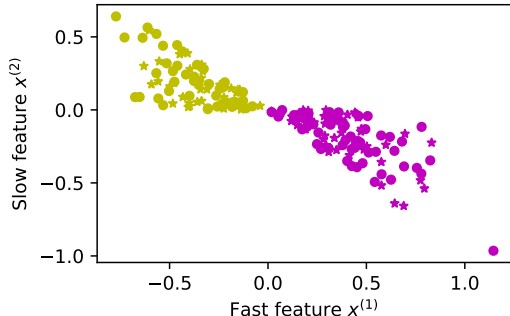

Figure 7: **Toy dataset (equi-distributional setting).** For this illustration, we take $\theta = \pi/4$ and $\varepsilon = 0.1$. Training data points are circles, while test data points are stars. The color corresponds to the data point's true label $y$.

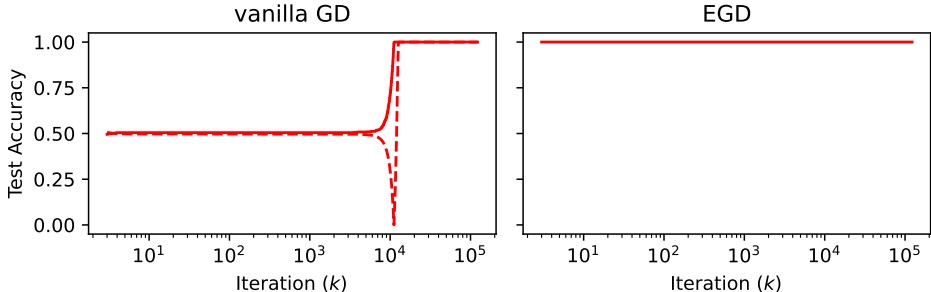

Figure 8: Empirical verification of Theorem 2. Solid lines are actual experiments, while lines correspond to theoretical predictions according to Eqn. (19).

**Population Moments (Exact 1D Representations).**   Define the population second-moment matrix and cross-moment vector:

$$\Sigma = \mathbb{E}_P[xx^\top] = \begin{pmatrix} a & b \\ b & d \end{pmatrix}, \qquad c = \mathbb{E}_P[yx] = \begin{pmatrix} c_1 \\ c_2 \end{pmatrix}, \qquad w_* = \Sigma^{-1}c = \begin{pmatrix} \mu_* \\ \nu_* \end{pmatrix}^\top.$$

Let $\kappa = \cot\theta/\sqrt{\varepsilon}$, $\phi$ and $\Phi$ the standard normal PDF and CDF. All moments admit absolutely convergent 1D representations:

$$Z = \mathbb{P}_P\big((x^\top e_1)(x^\top v) \le 0\big) = \frac{1}{\pi}\arccos\Big(\frac{\cos\theta}{\sqrt{\cos^2\theta + \varepsilon\sin^2\theta}}\Big),$$

$$a = \frac{2}{Z}\int_0^\infty t^2\,\phi(t)\,\Phi(\kappa t)\,dt, \quad d = \frac{2\varepsilon}{Z}\int_0^\infty\Big(1 - \frac{\kappa t\,\phi(\kappa t)}{\Phi(\kappa t)}\Big)\phi(t)\,\Phi(\kappa t)\,dt,$$

$$b = -\frac{2\sqrt{\varepsilon}}{Z}\int_0^\infty t\,\phi(t)\,\phi(\kappa t)\,dt, \quad c_1 = \frac{2}{Z}\int_0^\infty t\,\phi(t)\,\Phi(\kappa t)\,dt, \quad c_2 = -\frac{2\sqrt{\varepsilon}}{Z}\int_0^\infty \phi(t)\,\phi(\kappa t)\,dt.$$

With a bit of work, one can show that

$$\lambda_+ = \Theta(\varepsilon), \qquad \lambda_- = \Theta(\varepsilon), \qquad |b| = O(\sqrt{\varepsilon}).$$

**Vanilla Gradient Descent and Deterministic Equivalent.**   Vanilla GD on squared loss with fixed stepsize $\eta > 0$ (assuming $\eta < 1/\lambda_{\max}(\hat{\Sigma})$):

$$w(k+1) = w(k) - 2\eta\,(\hat{\Sigma}w(k) - \hat{c}), \qquad w(0) = (u_1, u_2)^\top,$$

where $\hat{\Sigma}$, $\hat{c}$, and $\hat{w}_*$ are empirical counterparts of $\Sigma$, $c$, and $w_*$ respectively, defined by

$$\hat{\Sigma} := \frac{1}{n}X^\top X = \frac{1}{n}\sum_{i=1}^n x_i x_i^\top, \quad \hat{c} := \frac{1}{n}X^\top Y = \frac{1}{n}\sum_{i=1}^n y_i x_i, \quad \hat{w}_* = w_{\text{OLS}} := \hat{\Sigma}^{-1}\hat{c}.$$

The *population recursion* (deterministic equivalent, exact in the large-$n$ limit) is

$$w(k+1) = w(k) - 2\eta\left(\Sigma w(k) - c\right), \qquad w(0) = (u_1, u_2)^\top.$$

Closed form:

$$w(k) = w_* - (I - 2\eta\Sigma)^k(w_* - w(0)).$$

Let $w(k) = (\mu_k, \nu_k)^\top$, $d_0 = w_* - w(0)$. Define

$$M = I - 2\eta\Sigma = \begin{pmatrix} 1 - 2\eta a & -2\eta b \\ -2\eta b & 1 - 2\eta d \end{pmatrix}, \qquad r_\pm = 1 - 2\eta\lambda_\pm \in (0, 1).$$

By Cayley–Hamilton Theorem,

$$M^k = \alpha_k I + \beta_k M, \qquad \alpha_k = \frac{r_+ r_-^k - r_- r_+^k}{r_+ - r_-}, \qquad \beta_k = \frac{r_+^k - r_-^k}{r_+ - r_-}.$$

we deduce that $\mu_k$ and $\nu_k$ are given by

$$\mu_k = \mu_* - \left[\alpha_k(\mu_* - u_1) + \beta_k\left((1 - 2\eta a)(\mu_* - u_1) - 2\eta b(\nu_* - u_2)\right)\right], \qquad (17)$$

$$\nu_k = \nu_* - \left[\alpha_k(\nu_* - u_2) + \beta_k\left(-2\eta b(\mu_* - u_1) + (1 - 2\eta d)(\nu_* - u_2)\right)\right]. \qquad (18)$$

**Exact Expression for Test Error.** Define the following scalars

$$\rho = \frac{\cos\theta}{\sqrt{\cos^2\theta + \varepsilon\sin^2\theta}}, \qquad \rho_k = \frac{\mu_k}{L_k}, \text{ with } L_k := \sqrt{\mu_k^2 + \varepsilon\nu_k^2}.$$

Working along the same lines as in the proof of Theorem 1, we obtain that the test error of $w(k)$ is given by

$$E_{test}\left(w(k)\right) := \mathbb{P}_{(x,y)\sim P}(yx^\top w(k) \le 0) = \frac{\arccos\left(\max(\rho, \rho_k)\right)}{\arccos(\rho)}. \qquad (19)$$

From the above analytical formula, we see that starts decreasing precisely when $\rho_k > \rho$, i.e.

$$|\nu_k| < |\mu_k||\tan\theta|.$$

For large $k \ge k_0 = O(1/\eta)$ the fast mode has already converged, so to leading order the drop occurs when

$$|\nu_k| \le T, \text{ where } T = T(\theta) := |\mu_*||\tan\theta|.$$

**Two-Sided Estimate of the Plateau Length.** Define the integer $k_* \in \mathbb{N} \cup \{\infty\}$ by

$$k_* := \min\{k \ge 0 : |\nu_k| \le T\}. \qquad (20)$$

The slow eigenvector is $q_- = e_2 + O(\sqrt{\varepsilon})$. There exist constants $\underline{C}(\theta), \overline{C}(\theta) \in [1, 1 + O(\sqrt{\varepsilon})]$ such that for all $k \ge k_0$,

$$\underline{C}^{-1}|\nu_* - r_-^k(\nu_* - u_2)| \le |\nu_k| \le \overline{C}|\nu_* - r_-^k(\nu_* - u_2)|, \qquad r_- = 1 - 2\eta\lambda_-.$$

Hence, when $|u_2 - \nu_*| > \overline{C}T$,

$$\boxed{\frac{1}{2\eta\lambda_-}\log\frac{|u_2 - \nu_*|}{\overline{C}T} \le k_* \le \frac{1}{2\eta\lambda_-}\log\frac{|u_2 - \nu_*|}{T/\underline{C}}.}$$

Since $\lambda_- = \Theta(\varepsilon)$, it follows that for $|u_2 - \nu_*| \ge T$,

$$\boxed{k_* \asymp \frac{1}{\eta\varepsilon}\log\frac{|u_2 - \nu_*|}{T} \quad \text{(up to multiplicative constants of order } 1 + O(\sqrt{\varepsilon})\text{).}}$$

Putting things together begets the following result.

**Theorem 2.** *If (i) $m_2(\theta) \le m_1(\theta)\tan\theta$ and (ii) $u_2 > \alpha u_1$, then the test error has a plateau of length $\Theta(\frac{1}{\eta\varepsilon})$. If either one of the conditions fails, then there is no plateau.*

Empirical verification of Theorem 2 is shown in Figure 8.

## B   Hyper-parameters for Experiments in Section 5

Hyper-parameters used in different tasks are listed in Table 1. In this table, $n$ is the number of input bits for the subset parity task, DR is data ratio which shows what fraction of all possible combinations have been used as the training set, LR is the learning rate, WD is weight decay, and BS is batch size. In all of the cases ReLU has been used as the activation function.

Table 1: Hyper-parameters for Different Tasks

| Task | Setting | $n$ | DR | Width | LR | WD | BS |
|:---:|:---:|:---:|:---:|:---:|:---:|:---:|:---:|
| | $k = 2$ | 400 | | 50 | 0.01 | $10^{-3}$ | |
| Subset Parity | $k = 3$ | 100 | NA | 100 | 0.042 | $10^{-2}$ | 32 |
| | $k = 4$ | 50 | | 100 | 0.023 | $10^{-2}$ | |
| | $p = 79$ | | | | | | |
| Modular Add./Mult. | $p = 97$ | NA | 0.5 | 512 | 0.7 | $10^{-4}$ | 512 |
| | $p = 127$ | | | | | | |

## C   Randomized SVD for Reducing Computational Cost

Performing SVD at each iteration introduces computational overhead. Although in some applications the total wall-clock time to reach high accuracy is not as important as the number of iterations (for example, in reinforcement learning it is important to minimize the number of iterations, which translates to interactions), to make EGD more plausible in practice we need to reduce the computational overhead of EGD. One way of doing this is to stop EGD at the epoch at which a desirable validation accuracy is reached. However, if we do not have access to a validation set, we need another method that reduces the per-iteration cost of SVD. Randomized SVD (**RSVD**) is a suitable candidate for doing that. The idea behind RSVD is that we can adopt principles from randomized linear algebra to down project a matrix into lower dimensional space, and then calculate SVD for the smaller matrix (Halko et al., 2011). We have shown the details of RSVD in Algorithm 3. Our implementation is a simplified version of the official scikit-learn scikit-learn developers (2024). As the gradient matrices are not square all SVD decompositions are performed in truncated form which has symmetric cost with respect to matrix dimensions.

Although EGD with exact SVD does not require tuning any hyper-parameter, RSVD has two important parameters: rank and number of iterations in the inner loop. We have set the number of iterations in the inner loop to 2 in all of the following experiments. In this section, we try to see what rank is the best for each task and what trade-offs we need to consider in tuning the rank. It should be noted that all of the execution times reported in this section have been calculated on the same Tesla T4 GPU, without any other tasks running and affecting the speed. Details on the experimental setup are provided in Section 5 and the hyper-parameters used is listed in Appendix B.

**Modular Addition.**   The first task we are going to study is the modular addition problem. Table 2 shows the number of epochs required to reach 95 percent accuracy, total wall-clock time to reach that accuracy, and the execution time per epoch for each method. As it can be seen, in all of the cases regular SVD outperforms all other methods in the sense of the number of epochs to reach the accuracy. However, when we consider the wall-clock time, in two out of three cases RSVD with rank 128 outperforms SVD. These results clearly show that if we care about the wall-clock time, there is an optimal rank that does not delay grokking too much while reducing per-epoch computational cost. Figure 9 shows how changing the rank of RSVD will change the epoch at which grokking happens. To compare column normalization, which is an approximation of EGD, with EGD and its randomized variants (i.e., EGD both with exact and randomized SVD, with different choices of the approximation rank), we see that column normalization achieves 95 percent accuracy later than almost all variants of EGD. However, in the sense of wall-clock time, in two out of three cases column normalization is the best method due to minimal computational overhead compared to vanilla SGD.

| p | Method | Epochs to 95% ↓ (↑) | Time to 95% (s) ↓ (↑) | Time/Epoch (ms) ↓ |
|---|---|---|---|---|
| | Vanilla SGD | 10099 | 133.3 | 13.2 |
| | EGD (SVD) | **222(45.5x)** | 12.5(10.6x) | 56.4 |
| 79 | EGD (RSVD, r=128) | 224(45.1x) | **11.1(12.0x)** | 49.7 |
| | EGD (RSVD, r=64) | 397(25.4x) | 14.0(9.5x) | 35.3 |
| | Column Norm. | 863(11.7x) | 15.4(8.6x) | 17.8 |
| | Vanilla SGD | 5392 | 156.9 | 29.1 |
| | EGD (SVD) | **116(46.5x)** | 10.9(14.4x) | 94.4 |
| 97 | EGD (RSVD, r=128) | 139(38.8x) | 11.5(13.6x) | 82.8 |
| | EGD (RSVD, r=64) | 237(22.7x) | 12.2(12.9x) | 51.4 |
| | Column Norm. | 328(16.4x) | **10.3(15.2x)** | 30.6 |
| | Vanilla SGD | 3293 | 123.8 | 37.6 |
| | EGD (SVD) | **62(53.1x)** | 11.4(10.8x) | 184.4 |
| 127 | EGD (RSVD, r=128) | 76(43.3x) | 10.1(12.2x) | 132.5 |
| | EGD (RSVD, r=64) | 146(22.5x) | 12.1(10.2x) | 83.2 |
| | Column Norm. | 133(24.7x) | **5.9(21.0x)** | 44.1 |

Table 2: **Results on Modular Addition.** Comparison of Vanilla SGD, EGD with exact SVD), EGD with randomized SVD (RSVD), and column normalization (a simplified version of EGD) on the modular addition problem, reporting epochs to 95% accuracy, wall-clock time (in seconds), and per-epoch execution time (in milliseconds). The numbers written in parentheses are the result of dividing the corresponding value for vanilla SGD by the entry. Therefore, larger numbers in parentheses mean being faster by the written factor compared to vanilla SGD. In all of the entries, a lower number outside the parentheses and a larger number in parentheses is better. In the sense of the number of epochs, EGD with exact SVD is the best. In the sense of the time to reach 95% accuracy, column normalization is the best, and EGD with RSVD and rank 128 is the second best on average.

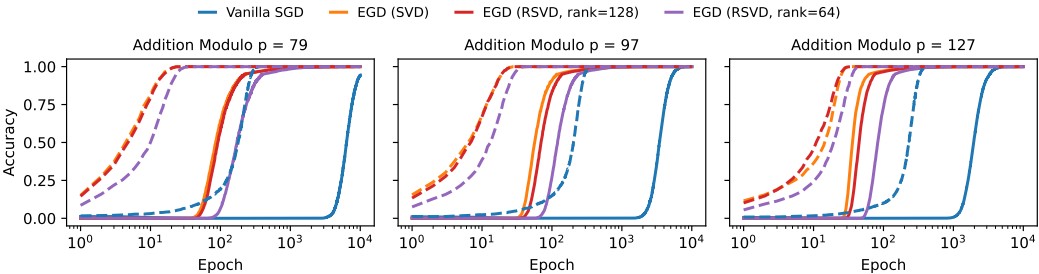

Figure 9: **Results on Modular Addition** for different values of the modulus p. Solid lines correspond to test accuracy and broken lines correspond to train accuracy. The plots show that using randomized-SVD and reducing its rank prevents EGD from reaching its full acceleration potential. However, when we consider the wall-clock time, randomized SVD with rank 128 seems to be the optimal rank, as it does not delay grokking as much as smaller ranks while increasing per-step speed. Refer to Section 5 for details on the experimental setup and to Appendix B for the hyper-parameters used.

**Modular Multiplication.** For this problem as well, we have compared EGD with exact SVD and its computationally more efficient variant based on randomized SVD (RSVD). Table 3 shows execution time results, and Figure 10 shows learning curves. The conclusion is the same as for modular addition. The simplified version of EGD, which is SVD-free (column normalization), achieves the best wall-clock time. However, if we care about both the number of epochs to reach 95% and the wall-clock time, EGD with randomized-SVD with rank 128 is the best overall method, as it reduces the wall-clock time while not significantly increasing the grokking epoch.

**Sparse Parity Problem.** For the sparse parity problem, we have tried 4 different ranks for RSVD to see the trade-offs more clearly. Interestingly, for all $k$ values, there is an RSVD setup which achieves 95% accuracy both in fewer epochs and in less wall-clock time compared to EGD with exact SVD. This gives us a clue that optimizing this rank is crucial in practice. The execution time results are listed in Table 4, and the learning curves are shown in Figure 11. For the sparse parity problem,

| p | Method | Epochs to 95% ↓ (↑) | Time to 95% (s) ↓ (↑) | Time/Epoch (ms) ↓ |
|---|---|---|---|---|
| | Vanilla SGD | 10116 | 133.5 | 13.2 |
| | EGD (SVD) | **221(45.8x)** | **11.4(11.7x)** | 51.7 |
| 79 | EGD (RSVD, r=128) | 233(43.4x) | 11.6(11.5x) | 49.8 |
| | EGD (RSVD, r=64) | 371(27.3x) | 12.9(10.3x) | 34.9 |
| | Column Normalization | 814(12.4x) | 13.8(9.7x) | 17.0 |
| | Vanilla SGD | 5715 | 115.4 | 20.2 |
| | EGD (SVD) | **107(53.4x)** | 10.4(11.1x) | 97.5 |
| 97 | EGD (RSVD, r=128) | 126(45.3x) | 9.9(11.6x) | 78.6 |
| | EGD (RSVD, r=64) | 233(24.5x) | 12.7(9.1x) | 54.7 |
| | Column Normalization | 328(17.4x) | **8.4(13.7x)** | 25.5 |
| | Vanilla SGD | 3261 | 99.8 | 30.6 |
| | EGD (SVD) | **65(50.2x)** | 11.9(8.4x) | 183.9 |
| 127 | EGD (RSVD, r=128) | 74(44.1x) | 9.8(10.2x) | 132.0 |
| | EGD (RSVD, r=64) | 142(23.0x) | 11.5(8.7x) | 80.8 |
| | Column Normalization | 142(23.0x) | **5.8(17.2x)** | 41.2 |

Table 3: **Results on Modular Multiplication.** Comparison of Vanilla SGD, EGD (with exact SVD), EGD (RSVD), and column normalization (a simplified version of EGD) on the modular multiplication problem, reporting epochs to 95% accuracy, wall-clock time (in seconds), and per-epoch execution time (in milliseconds). The numbers written in parentheses are the result of dividing the corresponding value for vanilla SGD by the entry. Therefore, larger numbers in parentheses mean being faster by the written factor compared to vanilla SGD. In all of the entries, a lower number outside the parentheses and a larger number in parentheses is better. In the sense of the number of epochs, EGD with exact SVD is the best. In the sense of the time to reach 95% accuracy, column normalization is the best, and EGD with randomized SVD (RSVD) and rank 128 is the second best on average.

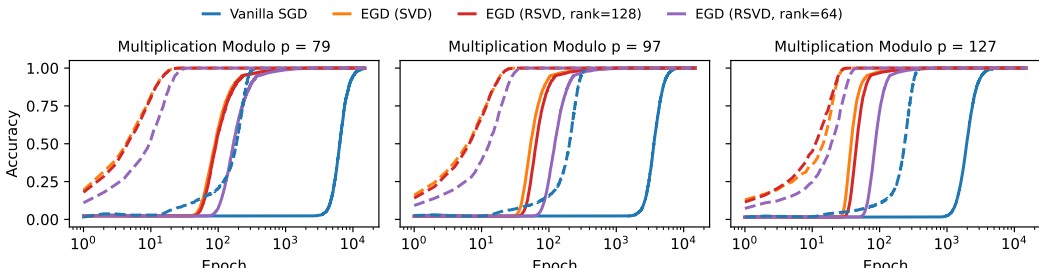

Figure 10: **Results on Modular Multiplication** for different values of the modulus $p$, for different values of $p$. Solid lines correspond to test accuracy and broken lines correspond to train accuracy. The same as the modular addition problem, in this problem also randomized SVD (RSVD) with rank 128 seems to be optimal, as it achieves faster per-epoch computation and minimally delays grokking compared to EGD with exact SVD. Refer to Section 5 for details on the experimental setup and to Appendix B for the hyper-parameters used.

column normalization is not as reliable, as for $k = 4$ it does not reach 95% accuracy, and in other cases, one of the randomized SVD variants with rank 40 is faster than this method both in terms of the number of epochs and wall-clock time.

**Average Over All Tasks.** In this section, for each problem we consider the best setup for each of the methods: EGD with exact SVD, EGD with randomized SVD (RSVD), and column normalization, and we take the average over the gains these provided compared to Vanilla SGD. Results are reported in Table 5. These results show that if we optimize the rank for all tasks, on average EGD with RSVD provides the best gain both in the number of epochs to achieve 95% accuracy and the wall-clock time.

| k | Method | Epochs to 95% ↓ (↑) | Time to 95% (s) ↓ (↑) | Time/Epoch (ms) ↓ |
|---|---|---|---|---|
| 2 | Vanilla SGD | 7873 | 409.4 | 52.0 |
| | EGD (SVD) | 109(72.2x) | 28.3(14.5x) | 259.5 |
| | EGD (RSVD, r=40) | **42(187.4x)** | **9.1(45.0x)** | 216.7 |
| | EGD (RSVD, r=30) | 424(18.6x) | 70.7(5.8x) | 166.7 |
| | EGD (RSVD, r=20) | 890(8.8x) | 137.5(3.0x) | 154.5 |
| | EGD (RSVD, r=10) | 2586(3.0x) | 354.5(1.15x) | 137.1 |
| | Column Normalization | 520(15.1x) | 40.4(10.1x) | 77.8 |
| 3 | Vanilla SGD | 1784 | 89.4 | 50.1 |
| | EGD (SVD) | 115(15.5x) | 36.3(2.5x) | 315.6 |
| | EGD (RSVD, r=40) | 180(9.9x) | 39.7(2.2x) | 220.7 |
| | EGD (RSVD, r=30) | **76(23.5x)** | 13.5(6.6x) | 178.0 |
| | EGD (RSVD, r=20) | 99(18.0x) | 15.2(5.9x) | 153.8 |
| | EGD (RSVD, r=10) | 312(5.7x) | 41.4(2.1x) | 132.6 |
| | Column Normalization | 120(14.9x) | **8.8(10.1x)** | 73.8 |
| 4 | Vanilla SGD | 5273 | 253.1 | 48.0 |
| | EGD (SVD) | 254(20.7x) | 50.5(5.0x) | 199.0 |
| | EGD (RSVD, r=40) | - | - | 183.0 |
| | EGD (RSVD, r=30) | **176(30.0x)** | **27.6(9.2x)** | 157.0 |
| | EGD (RSVD, r=20) | 312(16.9x) | 45.5(5.6x) | 145.9 |
| | EGD (RSVD, r=10) | 1067(4.9x) | 136.5(1.8x) | 127.9 |
| | Column Normalization | - | - | 69.9 |

Table 4: **Results on Sparse Parity Problem.** Comparison of Vanilla SGD, EGD (with exact SVD), EGD with randomized SVD (RSVD), and column normalization (a simplified version of EGD) on the modular multiplication problem, reporting epochs to 95time (in milliseconds). The numbers written in parentheses are the result of dividing the corresponding value for vanilla SGD by the entry. Therefore, larger numbers in parentheses mean being faster by the written factor compared to vanilla SGD. In all of the entries, a lower number outside the parentheses and a larger number in parentheses is better. Interestingly for this problem, in all three cases, there is an RSVD setup which leads to faster grokking compared to EGD with exact SVD. The best method both in the sense of number of epochs and wall-clock times is EGD with RSVD with rank 30.

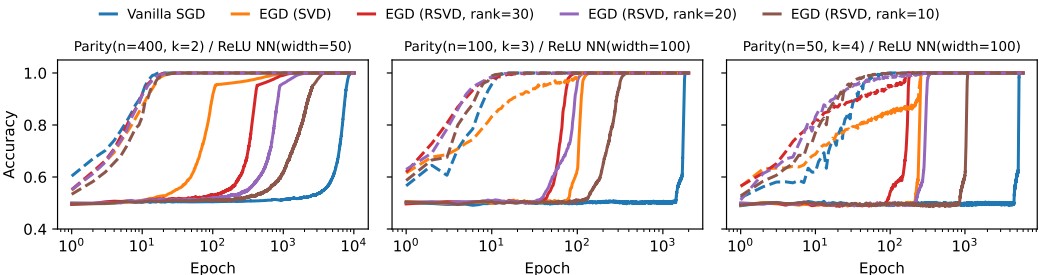

Figure 11: **Results on Sparse Parity Problem.** Solid lines correspond to test accuracy and broken lines correspond to train accuracy. Interestingly in this problem, there are some randomized SVD (RSVD) setups which grok in fewer epochs compared to EGD with exact SVD. This task shows that although column normalization is a successful light version of EGD, it is not reliable as it does not lead to an increase in test accuracy. Refer to section 5 for details on the experimental setup and to Appendix B for the hyper-parameters used.

# D    EGD IN MORE PRACTICAL SCENARIOS AND EMPIRICAL COMPARISON WITH GROKFAST

Grokking has been detected in learning more practical datasets, such as MNIST(LeCun et al., 2002), and in modular arithmetic tasks when using more complex architectures, such as a 2-layer transformer (Liu et al., 2023; Lee et al., 2024). In this section, we analyze whether EGD and EGD

| Method | Epochs to 95% ↑ | Time to 95% (s) ↑ |
|---|---|---|
| EGD (SVD) | 44.8x | 9.8x |
| EGD (RSVD, r=Best for each Task) | **55.6x** | **14.7x** |
| Column Normalization | 16.9x | 13.2x |

Table 5: Average of improvement gained by the best setup for each task. For Column Normalization, Sparse Parity Problem with $k = 4$ is excluded as the test accuracy does not increase at all in this case.

with randomized-SVD can accelerate grokking in these more practical scenarios as well. We also compare our proposed method to Grokfast, which is the strongest baseline available. We have used the same hyper-parameters and optimizer as reported in the Grokfast paper and used the official codebase provided by the authors to ensure a fair comparison. In all of the following comparisons, we have used the best Grokfast setup, which uses EMA to filter gradients instead of a memory bank to minimally add memory overhead to the vanilla optimizer. As discussed before, our method does not require any extra memory usage.

In this section, we show that EGD provides meaningful gains in more practical scenarios, when the dataset is more complex (MNIST), when the model is more complex than a 2-layer MLP (3-layer MLP and and 2-layer transformer), and when the optimizer is more complex than vanilla SGD (Adam and AdamW).

**MNIST Dataset with a 3-Layer MLP as the Model and AdamW as the Optimizer.** It has been shown that with some specific hyper-parameters, a 3-layer ReLU MLP network exhibits grokking when trained on the MNIST dataset (Liu et al., 2023). We have used the same parameters, optimizer (AdamW (Loshchilov & Hutter, 2017)), and model as reported by Omnigrok (Liu et al., 2023) and Grokfast (Lee et al., 2024). We perform EGD on the weights of all three layers of the MLP network. Results are summarized in Table 6. EGD and all of its variants based on randomized SVD (RSVD) achieve 85% accuracy in fewer optimization steps than Grokfast. However, EGD with SVD only minimally outperforms Grokfast in the sense of wall-clock time to reach 85% (85 percent is chosen to be consistent with the results reported by Grokfast), but the variants of EGD with RSVD outperform Grokfast by a large margin both in the number of steps and in time to reach 85%. If we look at the learning curves shown in Figures 12 and 13, we see that EGD and its variants grok faster than Grokfast. Also, in Grokfast there is a sudden drop and then increase, which we do not see in the EGD plots. Another interesting observation is that the training loss plots are noisy for both Vanilla AdamW and Grokfast. However, EGD has a smooth training loss curve.

| Method | Steps to 85% ↓ (↑) | Time to 85% (s) ↓ (↑) | Time/Epoch (ms) ↓ | Final Val. Acc. ↑ |
|---|---|---|---|---|
| Vanilla AdamW | 43100 | 660.0 | 15.3 | 89.0% |
| Grokfast | 1800(23.9x) | 28.7(23.0x) | 16.0 | 91.0% |
| EGD(SVD) | 700(61.6x) | 24.3(27.2x) | 34.7 | **91.9%** |
| EGD(RSVD,r=16) | 600(71.8x) | 11.7(56.4x) | 19.5 | 91.3% |
| EGD(RSVD,r=8) | **500(86.2x)** | 9.4(70.2x) | 18.8 | 89.6% |
| EGD(RSVD,r=4) | **500(86.2x)** | **9.2(71.7x)** | 18.4 | 91.5% |

Table 6: **Results on MNIST.** We have used the same parameters, model, and optimizer suggested by the Grokfast paper. As can be seen, RSVD-based variants of our proposed EGD outperform Grokfast both in the number of optimization steps and in the time to reach 85%. Also, in terms of final validation accuracy, EGD (both with exact and randomized SVD) variants outperform Grokfast.

**Modular Arithmetic with a 2-Layer Transformer as the Model and Adam as the Optimizer.** Following the setup suggested and used by previous papers (Liu et al., 2023; Lee et al., 2024), we use a 2-layer transformer and the Adam optimizer (Adam et al., 2014) as the baseline to compare with EGD. We also compare results with Grokfast (Lee et al., 2024). The EGD is only performed on the first MLP layer of transformer. Execution time results are listed in Table 7. The results suggest that EGD with RSVD performs comparably to Grokfast. However, our method does not add any overhead to memory. These results are based on comparing EGD with Grokfast, which uses EMA for filtering. However, to calculate EMA, Grokfast requires memorizing a copy of gradients to update. In EGD, no extra memory is needed. The learning curve comparisons are shown in Figures 14 and

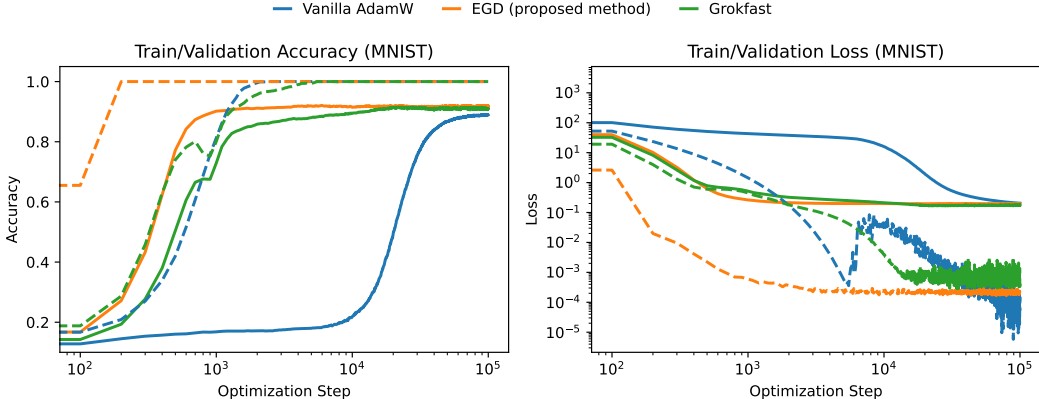

Figure 12: **Results on MNIST.** Solid lines correspond to validation accuracy/loss and broken lines correspond to train accuracy/loss. Here our EGD is implemented via exact SVD. As it can be seen, EGD accelerates grokking more than Grokfast. Also, it does not show any sudden decrease in accuracies or large noise in train loss.

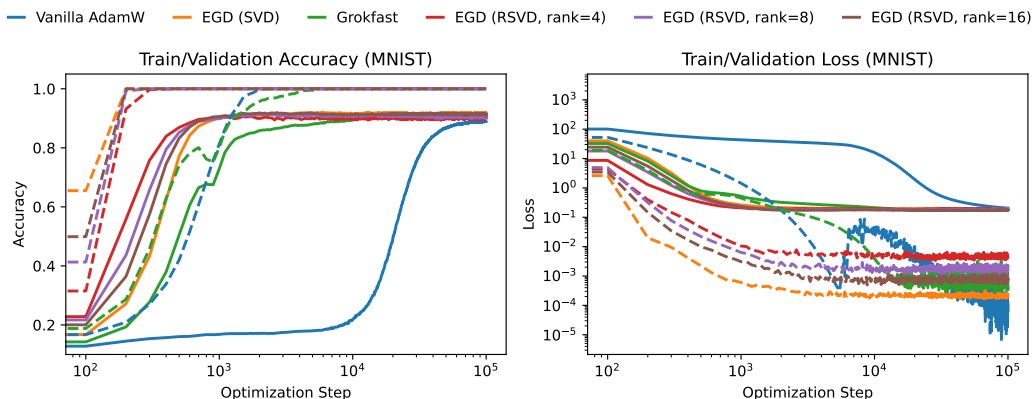

Figure 13: **Results on MNIST.** Solid lines correspond to validation accuracy/loss and broken lines correspond to train accuracy/loss. Here, our EGD method is implemented via exact SVD and randomized SVD (RSVD) with different ranks. We see that for some ranks, RSVD groks even faster than EGD. All of the RSVD versions have smooth training los curvae in contrast with Grokfast and vanilla AdamW.

15. These curves show that EGD is extremely unstable in this scenario. However, when we use randomized-SVD, the results become stable and even more stable than Grokfast.

| Method | Steps to 95% ↓ (↑) | Time to 95% (s) ↓ (↑) | Time/Step (ms) ↓ | Final Val. Acc. ↑ |
|---|---|---|---|---|
| Vanilla Adam | 55780 | 5198.7 | 93.2 | 99.1% |
| Grokfast | **910(61.3x)** | **114.0(45.6x)** | 125.3 | 100% |
| EGD(SVD) | 1170(47.7x) | 179.9(28.9x) | 153.8 | 100% |
| EGD(RSVD,r=36) | 980(56.9x) | 129.3(40.2x) | 131.9 | 100% |

Table 7: **Results on Mudolo Multiplication** with $p = 97$ and a 2-layer transformer as the model. Grokfast outperforms EGD with exact SVD, both in the number of epochs and in the time to reach 95% accuracy. However, the difference between the RSVD case and Grokfast is negligible, while EGD with RSVD does not introduce any extra memory usage, which is crucial in large models such as transformers.

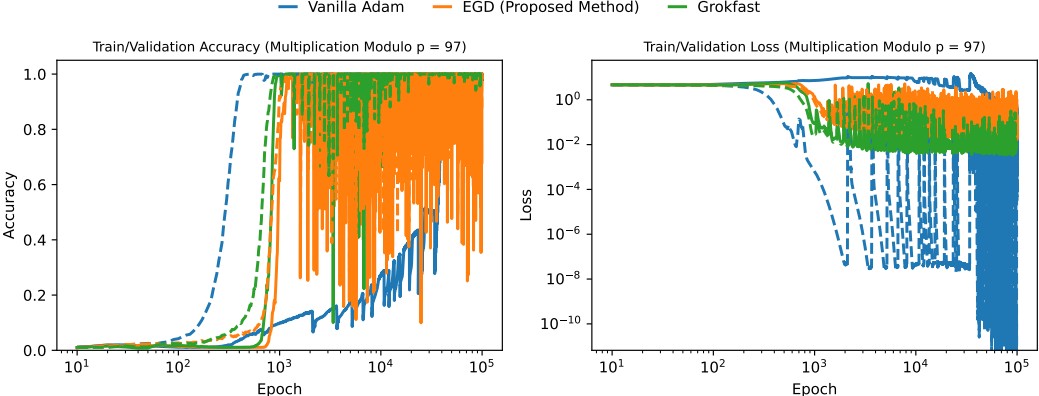

Figure 14: **Results on Mudolo Multiplication** for p=97. Solid lines correspond to validation accuracy/loss and broken lines correspond to train accuracy/loss. Here, our EGD method is implemented via SVD, and leads to unstable training for this task and model architecture.

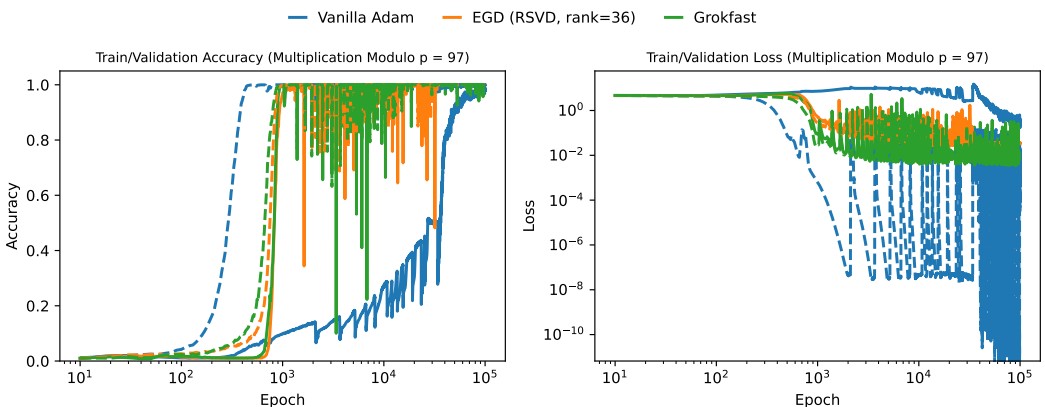

Figure 15: **Results on Mudolo Multiplication** for $p = 97$. Solid lines correspond to validation accuracy/loss and broken lines correspond to train accuracy/loss. Here, our EGD method is implemented via randomized SVD (RSVD) with rank = 36. Interestingly, EGD with RSVD here leads to more stable training, compared to Grokfast or EGD with exact SVD (Figure 14).

**Summary.** In practical scenarios, EGD with randomized SVD (RSVD) leads to faster grokking both in the number of steps and in wall-clock time without adding any extra memory utilization, which is crucial for large models and optimizers that already use a lot of memory to track momentum and velocity of gradients. However, we need to tune the rank of RSVD to achieve the best gain. The results are promising across various tasks, optimizers, and architectures.

## E   EGD HELPS WIDELY USED OPTIMIZERS HANDLE NON-STATIONARITY

In many real-world scenarios, handling non-stationarity is crucial. We hypothesize that our proposed method EGD, which leads to faster grokking and faster learning of nuances in data, will help optimizers adapt quickly to changes in the distribution of data. In this section, we test how adding EGD on top of three commonly used optimizers can improve their adaptability to changes in the distribution of data. We follow the experiment introduced by Sokar et al. (2023) to simulate non-stationarity. The idea is to start training on CIFAR-10 (Krizhevsky et al., 2009) and, at fixed intervals, shuffle all of the labels to see how fast the optimizer can recover and increase the training accuracy after shuffling. This experiment has been shown to be effective in demonstrating non-stationarity in reinforcement learning (Castanyer et al., 2025).

We have used the same setup and parameters used by Castanyer et al. (2025) for these experiments. The model is a Convolutional Neural Network with 3 convolution layers, max pooling, a fully connected layer, and a classification head. We perform EGD with SVD on all convolution layers and the fully connected layer. To make 3-dimensional convolution layers 2-dimensional, we flatten the last two dimensions. Figure 16 shows the effect of adding EGD on top of three commonly used optimizers (Adam (Adam et al., 2014), RAdam (Liu et al., 2019), and RMSprop (Tieleman & Hinton, 2017)) to enable them to recover faster after shuffling. In all cases, EGD increases the area under the curve. An interesting observation is that, without EGD, RMSprop seems to be more successful compared to the other two optimizers in handling non-stationarity. Table 8 shows that adding EGD improves handling of non-stationarity for all optimizers. The most significant improvement is observed for the Adam optimizer and when there are 3 points of shuffling.

| Number of Shuffling | Method | Adam | RAdam | RMSProp |
|---|---|---|---|---|
| Shuffling at 1 Epoch | Vanilla Optimizer | 43.0 | 40.8 | 60.2 |
| | Optimizer + EGD | 79.2(1.8x) | 70.0(1.7x) | 79.5(1.3x) |
| Shuffling at 3 Epochs | Vanilla Optimizer | 24.7 | 22.9 | 31.6 |
| | Optimizer + EGD | 56.0(2.3x) | 47.3(2.1x) | 54.7(1.7x) |
| Shuffling at 4 Epochs | Vanilla Optimizer | 23.0 | 19.8 | 40.1 |
| | Optimizer + EGD | 44.0(1.9x) | 23.5(1.2x) | 45.6(1.1x) |
| Shuffling at 9 Epochs | Vanilla Optimizer | 14.2 | 13.3 | 14.8 |
| | Optimizer + EGD | 24.3(1.7x) | 22.0(1.6x) | 24.0(1.6x) |

Table 8: **Non-Stationarity Injection Results on CIFAR-10.** Area under the curve for various optimizers after being subjected to a shift in distributions, with and without EGD. The gain from using EGD is most significant for Adam and RAdam, and the largest gain occurs when there are 3 points for shuffling labels.

## F DOES EGD CONVERGE TO DIFFERENT SUP-SPACE COMPARED TO VANILLA SGD?

Methods accelerating grokking usually do not converge to the same optimum as vanilla SGD (Lee et al., 2024). To test if that is the case for EGD as well, we have plotted the trajectory of weights while learning arithmetic tasks. The experimental details are the same as what provided in Section 5 and the hyper-parameters are listed in Appendix B. Figures 17, 19, and 21 show the trajectories for Addition, Multiplication, and Sparse Parity problems, respectively, and Figures 18, 20, 22 show the distance between weights at different time steps and the initialization. The following interesting observations can be made from these plots:

- Common across all tasks: after the initial big steps, EGD does not move much, but vanilla SGD moves over a wider range.

- Modulo problems: EGD and vanilla SGD take different and diverging trajectories. Both overshoot in distance from initialization. However, the distance is reduced afterwards.

- Sparse parity: although initially EGD and SGD trajectories diverge, SGD tries to reach the vicinity of the point at which EGD starts grokking. In terms of distance, both methods show an initial big jump and then maintain an almost fixed distance from the initialization.

Based on these observations, we can say that, except for the sparse parity problem, in the other two tasks EGD and vanilla SGD converge to different regions of the parameter space. In the sparse parity problem, they first diverge, and then vanilla SGD attempts to move toward the region where grokking occurs (i.e., vanilla SGD tries to turn toward the epoch at which EGD makes its large jump). The reason behind this behavior, and whether this observation also holds in the randomized case and for other tasks, is left as future work.

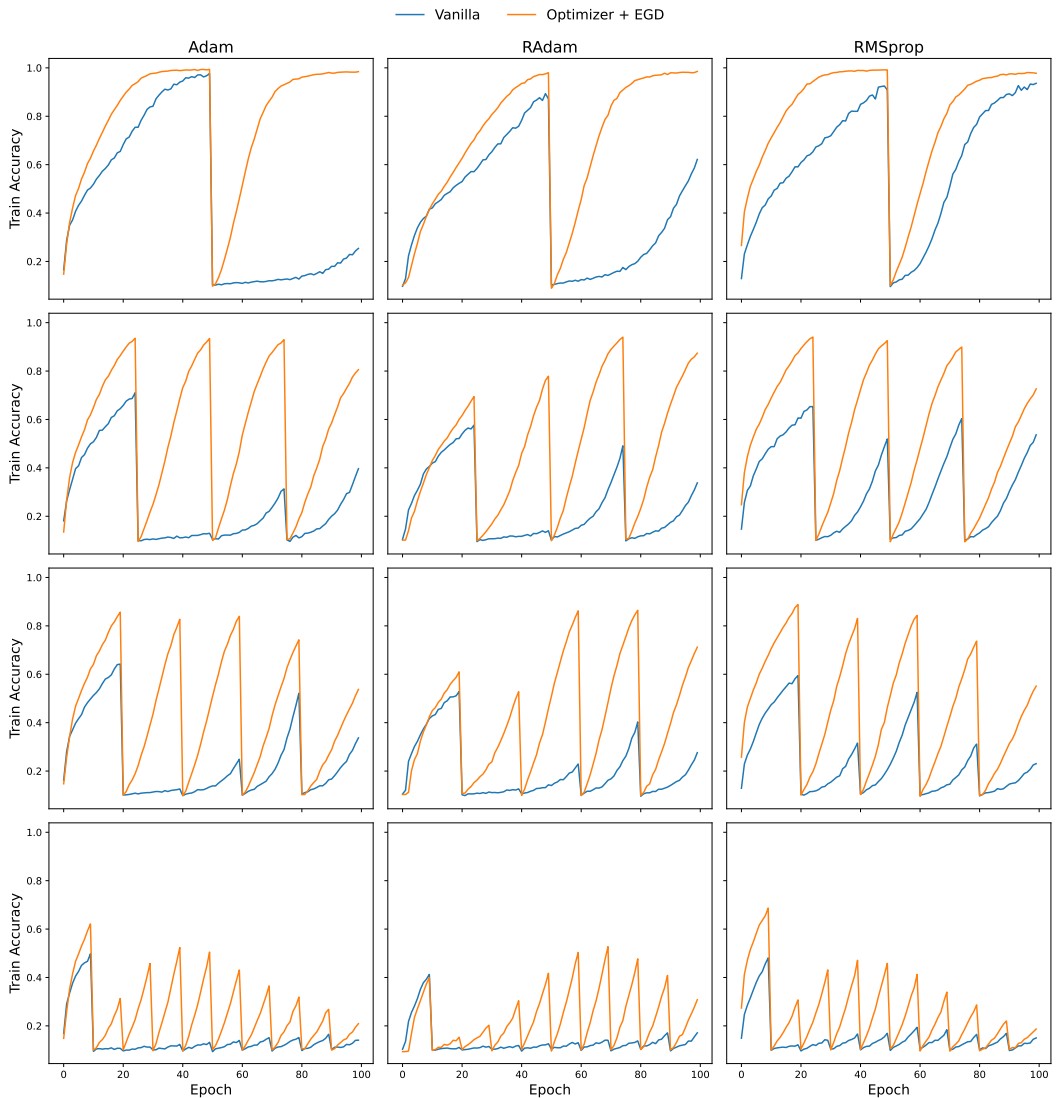

Figure 16: **Non-Stationarity Injection Results on CIFAR-10.** In each row, we shuffle the labels at the same intervals. From top to bottom, the number of shuffling points increases, and therefore recovery becomes more difficult. The results show that RMSprop is more successful in recovering after a shift in distribution. Also, in all optimizers and scenarios, adding EGD improved adaptability.

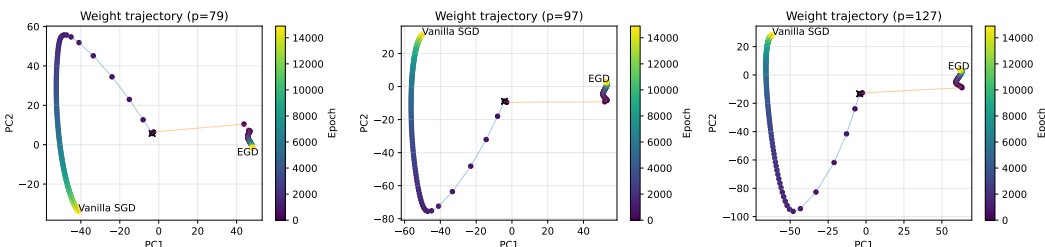

Figure 17: **Results on Modular Addition.** Trajectory during EGD and Vanilla SGD. EGD and SGD converge to different sub-spaces.

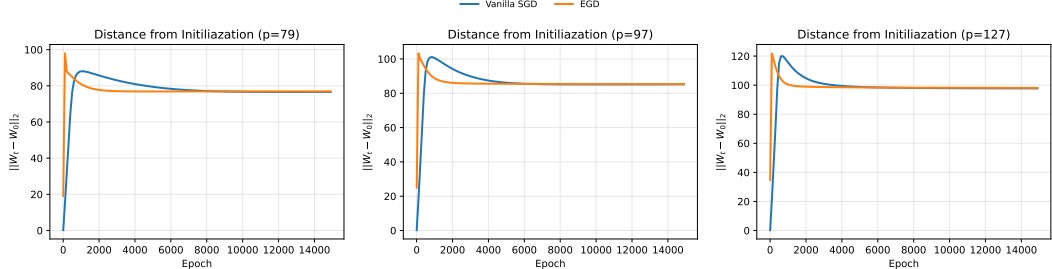

Figure 18: **Results on Modular Addition.** Distance between weights and the initializations. Both methods overshoot, and in the end converge to regions with the same and fixed distance from the initialization.

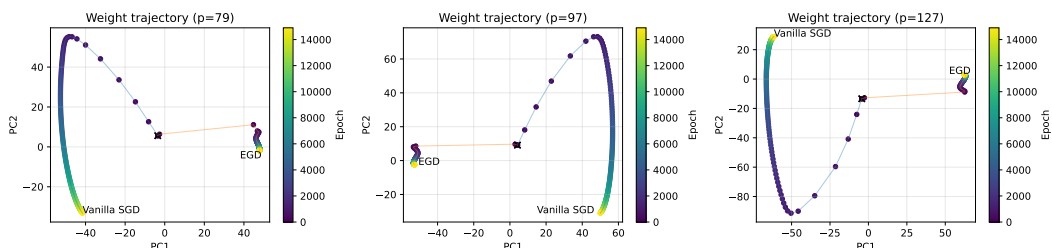

Figure 19: **Results on Modular Multiplication.** Trajectory during EGD and Vanilla SGD. EGD and SGD converge to different sub-spaces.

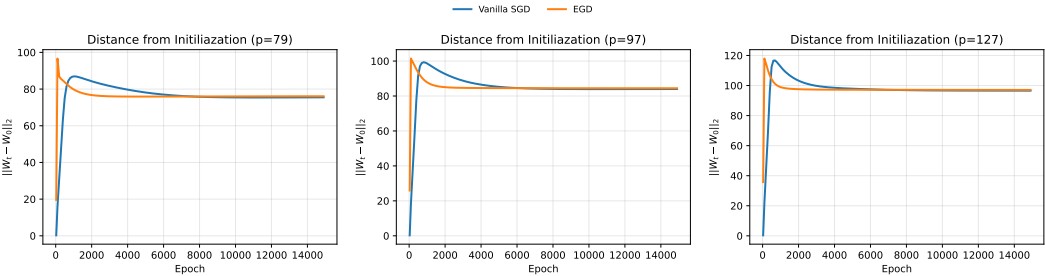

Figure 20: **Results on Modular Multiplication.** Distance between weights and the initializations. Both methods overshoot, and in the end converge to regions with the same and fixed distance from the initialization.

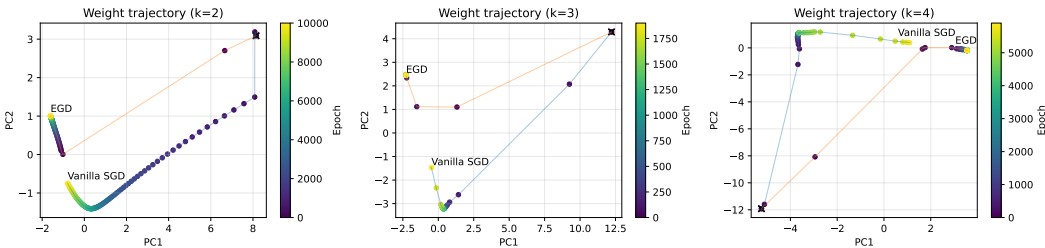

Figure 21: **Results on Sparse Parity Problem.** Trajectory during EGD and Vanilla SGD. EGD and SGD first diverge, then EGD turns towards the epoch at which EGD grokked.

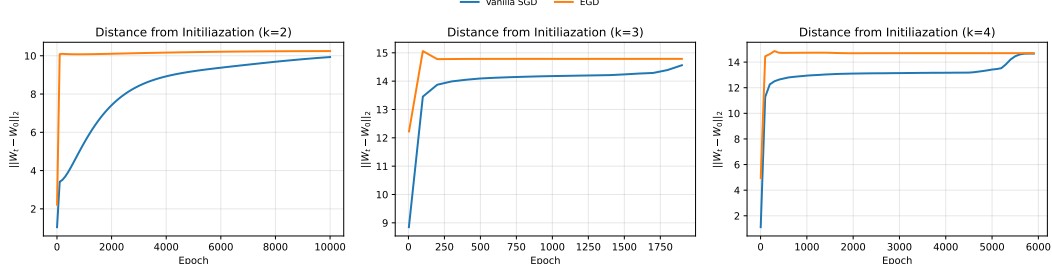

Figure 22: **Results on Sparse Parity Problem.** Distance between weights and the initializations. EGD overshoots first, but in the end remains in the same distance from initialization. Vanilla SGD does not overshoot and gradually goes to the region with the same distance to initialization as EGD.

## G    IMPLEMENTATION DETAILS AND PSEUDO-CODES

In this section, we provide the pseudo-code based on which we obtained the results in this paper. We present detailed algorithms for the three main components of our method. The descriptions of these algorithms are as follows:

- **Training Loop with EGD 1:** This box shows the general training loop with EGD. To reduce the cost, if we have access to a validation set and a threshold for acceptable validation accuracy, we stop using EGD after the epoch at which we attain that validation accuracy. Also, we can choose between RSVD and SVD.

- **EGD with Optional RSVD 2:** This algorithm accepts a gradient matrix and performs preconditioning based on SVD or randomized SVD. To get a symmetric cost with respect to dimensions of the matrix, we always calculate truncated SVD.

- **Randomized-SVD 3:** A detailed implementation of randomized SVD is shown in this box. We used a simplified version of the algorithm from the scikit-learn implementation (scikit-learn developers, 2024). The main difference between this algorithm and the official version is that we are not using oversampling. Studying the effect of oversampling is left to future work.

---

**Algorithm 1** TRAINING LOOP WITH EGD

**Require:**  parameters $\theta$, learning rate $\eta$, number of epochs $N_{\text{epoch}}$, training data $D_{\text{train}}$, validation data $D_{\text{val}}$, threshold $\lambda$, flag use_RSVD
1: **begin:**
    use_EGD $\leftarrow$ True
2: **for** epoch $= 1 \ldots N_{\text{epoch}}$ **do**
3:     **for** each batch $(X, y)$ in $D_{\text{train}}$ **do**
        Compute gradients: $G \leftarrow \nabla_\theta \mathcal{L}(\theta; X, y)$
4:         **if** use_EGD **then**
          $\tilde{G} \leftarrow$ EGD$(G, \text{use\_RSVD})$         EGD preconditioning (Algorithm 2 )
5:         **else**
          $\tilde{G} \leftarrow G$         No EGD applied
6:         **end if**
        Update parameters: $\theta \leftarrow \text{OptimizerStep}(\theta, \tilde{G}, \eta)$
7:     **end for**
    val_acc $\leftarrow$ Evaluate$(\theta, D_{\text{val}})$
8:     **if** val_acc $> \lambda$ **then**
        use_EGD $\leftarrow$ False         Stop EGD updates after threshold is reached
9:     **end if**
10: **end for**
    **Return:** trained parameters $\theta$

---

---

**Algorithm 2** EGD WITH OPTIONAL RSVD

---

1: **Input:** gradient matrix $G$ of size $d_{\text{out}} \times d_{\text{in}}$, flag use_RSVD
2: **begin:**
3: **if** use_RSVD **then**
4: $\quad (U, S, V^\top) \leftarrow \text{RSVD}(G)$
5: **else**
6: $\quad (U, S, V^\top) \leftarrow \text{SVD}(G)$ Truncated SVD
7: **end if**
8: $P \leftarrow U S^{-1} U^\top$ Construct pre-conditioning matrix
9: $\tilde{G} \leftarrow PG$
10: **Return:** $\tilde{G}$

---

---

**Algorithm 3** RANDOMIZED-SVD

---

1: **Input:** matrix $G$ of size $m \times n$, target rank $r$, iterations $n_{\text{iter}}$, seed
2: **begin:**
3: $m, n \leftarrow \text{shape}(G)$
4: $Q \leftarrow \text{Normal}(n, r, seed)$ Sample a Matrix with Random Gaussian entries
5: **for** $i = 1 \ldots n_{\text{iter}}$ **do**
6: $\quad Q \leftarrow \text{QR}(GQ)$ QR Decomposition
7: $\quad Q \leftarrow \text{QR}(G^T Q)$ QR Decomposition
8: **end for**
9: $B \leftarrow GQ$ Project onto low-rank subspace
10: $(\hat{U}, \hat{S}, \hat{V}^\top) \leftarrow \text{SVD}(B)$ Truncated SVD on small matrix $B$
11: **Return:** $\hat{U}, \hat{S}$ Return approximate singular vectors and values

---

