# OpenReview forum: "Egalitarian Gradient Descent: A Simple Approach to Accelerated Grokking"
_ICLR.cc/2026/Conference — ICLR 2026 Poster_

### Official Review · Reviewer_yxV7 · 2025-10-27

**Soundness:** 3
**Presentation:** 3
**Contribution:** 3
**Rating:** 4
**Confidence:** 3

**Summary:**

This paper studies the phenomenon of grokking—where neural networks plateau for long periods in generalization performance before abruptly improving—by connecting the length of the plateau to the spectral properties of the gradient during training. The authors propose Egalitarian Gradient Descent (EGD), a modification to classic gradient descent that normalizes the gradient along all principal directions, theoretically eliminating slow convergence due to ill-conditioned spectra. Extensive empirical evaluations are provided across toy problems and algorithmic tasks, demonstrating that EGD can dramatically reduce (or eliminate) the grokking plateau compared to vanilla SGD and other methods.

**Strengths:**

1. Clear theoretical insight and connection to spectrum: The paper gives a transparent analytic treatment for a simplified grokking scenario, precisely quantifying how ill-conditioning in the gradient’s principal directions translates into stagnation in test accuracy (see Section 3; Theorem 1 and Corollary 1). This directly links observable grokking delays to quantifiable optimization issues, moving beyond anecdotal claims.

 2. Simple, principled method: EGD is conceptually straightforward, introduces no free hyperparameters, is optimizer-agnostic, and is presented as a direct, closed-form modification to the gradient for each layer (Section 4). Compared to competing approaches like Grokfast, EGD incorporates no history/memory buffers, making it lightweight for integration.

 3. Sound mathematical formalism: Derivations (e.g., equations (11)-(12), as well as the detailed proofs in Appendix A) are logically consistent, and the singular-value “flattening” interpretation of the update is rigorous.

 4. Strong empirical backing: Results in Figures 1–3 show that EGD consistently accelerates grokking across modular addition, modular multiplication, and sparse parity datasets—tasks where the phenomenon has been robustly documented. EGD typically leads to immediate generalization, while SGD and even “column normalization” lag—this is evident in Figure 1 and Figure 2.

 5. Insightful spectral diagnostics: Figure 4 explicitly illustrates how delays in grokking correspond to highly skewed gradient spectra, making the theoretical contribution tangible and informative for the reader.

**Weaknesses:**

1. Scalability and Computational Cost: The method requires per-step SVD computation on the gradient matrix for each layer, an operation that grows rapidly with layer width. While the authors claim (Page 7) that randomized or approximate SVDs “work just fine”, there is no benchmark or ablation quantifying the tradeoff, or on what network scales/width SVD ceases to be efficient. This undermines claims about practical usability in modern architectures.

2. Lack of Ablation and Sensitivity Analysis: The experimental section has limited sensitivity checks. For example, EGD is briefly compared to a “column normalization” heuristic, but not to common natural gradient or second-order optimizers (despite Section 4.1’s close connection). There is no ablation quantifying: (i) the effect of running EGD on only top-$k$ directions vs. full-rank, or (ii) across various batch sizes/optimizers. This makes it hard to parse where the real gains come from.

3. Missing Rigorous Quantitative Comparisons with Strong Baselines: Grokfast (Lee et al., 2024) and other recent grokking accelerators are discussed theoretically (Section 4.2), but quantitative experimental comparisons are limited. The paper could benefit from more comprehensive results tables directly benchmarking EGD against these methods across identical tasks.

4. Overstated Generality and Claims Beyond the Evidence: In several places (e.g., Section 6, Conclusion; Abstract; and throughout the Introduction), the paper implies EGD could be universally beneficial and “plug-and-play” for any neural network. The only evidence is for the tightly controlled algorithmic family, and the real computational cost is glossed over. The claims regarding optimizer-agnosticism and hyperparameter-freeness require much more stress-testing.

5. Limited Theoretical Depth on Nonlinear/Deep Networks: Section 4 describes the transition from linear to nonlinear settings, but almost all the theoretical analysis is confined to the analytically tractable linear case, or at best layer-wise argument. No generalization, convergence, or stability claim is rigorously proven for deep, nonlinear architectures—the regime of real interest.

**Questions:**

1. Applicability/Scalability Beyond Toy Tasks: What is the maximal network width, depth, and dataset size on which you have successfully deployed EGD, and how does the runtime/throughput compare with vanilla SGD and other strong baselines? Can you provide concrete experiments or ablations addressing actual wall-clock cost?

 2. Necessity of Full-Rank SVD: Is the full SVD required for each gradient step in all layers, or would it suffice to approximate just the top-$k$ singular values/vectors, perhaps by sketching methods? (Experiments addressing this could open EGD to wider scaling.)

 3. Direct Empirical Comparison to Grokfast: Your only comparative discussion with Grokfast is qualitative. Can you provide synchronized benchmarks (same datasets, architectures, and hyperparameters) for EGD and Grokfast, so the empirical strengths and limitations are evident?

 4. Failure Modes or Adverse Side Effects: Can EGD cause instability, exploding/vanishing gradients, or harm convergence in high-noise regimes or complicated real-world architectures? Any evidence of problems in practical runs would be useful.

 5. Potential for Generalization Analysis in Deep Networks: Is there any theory in progress on whether the spectral-flattening argument for “egalitarian” updates extends robustly to multi-layer, highly nonlinear settings?

---

> ### Author Response · Authors · 2025-11-19
> **Response to Reviewer yxV7**
>
> We thank the reviewer for their great insights. We shall address the main remarks and questions of the reviewer in detail.
>
> ***Scalability, Computational Cost, and Going Beyond Arithmetic Setups.*** In Appendices D and E of the updated manuscript, we have added experiments showing successful usage of EGD on more realistic datasets (MNIST in Appendix D, and CIFAR-10 in Appendix E) and larger models (up to 3-layer MLPs in Appendix D, a 2-layer transformer in Appendix D, and 3-layer CNN in Appendix E). These experiments compare our proposed EGD (both with exact SVD and RSVD, i.e. randomized SVD, a very efficient low-rank approximation) versus Grokfast (Appendix D) and show EGD can successfully be added to widely used optimizers (Adam, RAdam, AdamW, and RMSprop) and makes them more adaptive (Appendices D and E). In our experiments, our proposed method accelerated grokking in all of the scenarios tested. Also, RSVD showed to be effective in reducing the per iteration cost while giving similar acceleration as SVD. This leads to much better wall-clock time.
>
> ***Ablation and Sensitivity Analysis.*** We have added Appendix C to show how randomized-SVD can be used to reduce the computational cost of EGD. The results of using RSVD are shown in Tables 2,3, and 4. Also learning curves are shown in Figures 7, 8, and 9. Also, in more practical scenarios, we have used RSVD as well (Appendix D). The interesting observation is that for the sparse parity problem, RSVD leads to the best grokking acceleration. On average, RSVD performs better than SVD both in epochs to reach 95 percent and time to reach that.
>
> ***Failure Modes, Adverse Effects, and Comparison to Grokfast.*** As mentioned earlier above, we have also added a comparison with Grokfast Appendix D. On MNIST, our proposed EGD method (with RSVD) outperforms Grokfast, in terms of faster grokking times and more stable train curves. For arithmetic problems with transformers as the model, EGD with exact SVD is unstable, but EGD with RSVD is comparable with Grokfast without requiring more memory footprint of the latter. Refer to Tables 7 and 6, alongside Figures, 10, 11, 12, and 13.
>
> ***Potential for Generalization Analysis in Deep Networks.*** Our experiments seem to indicate that for larger more complex models such as transformers, our proposed EGD method with RSVD is more stable than with exact SVD. Refer to Figures 12 and 13. However, in non-stationary situations EGD can be added to any optimizer and enables it to handle non-stationarity Appendix E (Table 8 and Figure 14). This is very promising and will be explored further in future work.
>
> We are address any further concerns the reviewer might still have.

---

> > ### Author Response · Authors · 2025-11-28
> > **Gentle reminder: Discussion period ends soon**
> >
> > Once again we thank the reviewer for their  time and useful insights on our work.
> >
> > Our response above (posed on November 19), addresses all the points raises by the reviewer, including additional experiments and ablation studies  on different models (CNN, transformers) and datasets (mnist, cifar), in response to the questions of the reviewer. We updated manuscript also includes extensive results (and discussions) on randomized SVD variant of our proposed EGD model.
> >
> > Since the discussion deadline is fast approaching, we would like invite the reviewer to consider our rebuttal. We are happy to address any further questions the reviewer might have.

---

### Official Review · Reviewer_vvi7 · 2025-10-30

**Soundness:** 3
**Presentation:** 3
**Contribution:** 3
**Rating:** 8
**Confidence:** 3

**Summary:**

This paper introduces Egalitarian Gradient Descent (EGD) as an optimization to reduce and or mitigate the grokking phenomenon. Theoretical rationale is provided as to why grokking occurs within training practices using stochastic gradient descent (SGD). They show that the leading singular directions dominate the optimisation process, which they show plays a role in grokking. Therefore, they propose the EGD, which sets all the singular directions of the gradient update to 1; thus, all singular directions are optimised equally, hence the name Egalitarian. They highlight that this method can be viewed as a simplified version of Grokfast and enjoys the same benefits while not requiring hyperparameter sweeps. The results of the method are shown on the sparse party problem and modular arithmetic tasks.

**Strengths:**

- The paper is very well written, reads well, and is self-contained.

- It is positioned very well within the literature exploring grokking and methods to mitigate/reduce grokking.

- It provides a strong theoretical rationale behind the method presented.

- Provides insights into the occurrence of grokking more generally.

**Weaknesses:**

- In Figures 1 and 2, it is not clear where the train accuracy lines are for the EGD method. Is this due to them going up in tandem, or has this data been omitted?

- Figures 1 and 2 range from -0.5 to 1, resulting in a large portion of the figures being clear; this should be reduced to make the data easier to read. It would also be useful to provide the grid to make reading off the figures easier.

- Increasing the number of tasked benchmarked would increase the impactfulness to the overall community, especially tasks where grokking is induced, such as on MNIST in [1].

- Although the method is fairly simple, as it requires performing SVD on the gradient to receive $U$ and $V^T$, it would be good to provide the codebase to enable the work to be easily replicated.

- The `Limitations and Future Work` section reads more like future work. The limitations of the work are not clearly represented. This should be more aptly named Future Work.

- The Bibliography is missing links to papers.

- Typo/Spelling Mistake: Line 375 occured should be `occurred`


[1] Liu, Z., Michaud, E.J. and Tegmark, M., 2022. Omnigrok: Grokking beyond algorithmic data. arXiv preprint arXiv:2210.01117.

**Questions:**

1.  Line `374` states In practice, we turn off EGD and switch it for vanilla (S)GD once we detect grokking has occured $^1$ where the footnote states This is detected by monitoring validation loss.  When does one exactly turn off EGD? At what point do you consider grokking detected? Is it once the model starts to generalise? Does this not introduce a hyperparameter?

2. For the sparsity parity problem, Figure 3. The method does not fully resolve grokking for Parity(n=100, k=3) ReLU NN(width=100). Is there a rationale behind why here the model training and test accuracy do not increase in tandem, as was shown in the modular arithmetic tasks?

3. Can you provide the results of this method on Grokking-Induced MNIST and IMBD dataset as done in [1]

[1] Liu, Z., Michaud, E.J. and Tegmark, M., 2022. Omnigrok: Grokking beyond algorithmic data. arXiv preprint arXiv:2210.01117.

---

> ### Author Response · Authors · 2025-11-19
> **Response to Review vvi7**
>
> We thank the reviewer for their great insights. We shall address the main remarks and questions of the reviewer in detail.
>
> ***Reframing “Limitations and Future Work”.*** We agree with the reviewers suggestion to rename that paragraph aptly to just “Future Work”. Please refer to the updated manuscript (all modifications appear in blue).
>
> ***Axis and Range of Figures 1 and 2.*** As we had large evaluation intervals, and the plots were in log-scale, the details of the beginning part of the plots were not clear and obvious. We decreased the evaluation interval to have access to all epochs during training and figures are updated in the paper now (Figure 1 and 2). Now these new figures show better the significant acceleration caused by using EGD (comparing orange and blue lines. The green line is column normalization which is  a simplification of EGD).
>
> ***When does one exactly turn off EGD?*** EGD stops at the epoch or iteration at which the validation accuracy reaches an acceptable threshold. We may not consider this threshold as a hyper-paramater as we do not need to tune it. In other words, we can stop it at any epoch we desire based on the accuracy we are looking for. However, if the validation dataset is not available, we have shown that randomized SVD (RSVD) is a strong alternative which can reduce total computation time (See Appendices C and D for details).
>
> ***Concerning Figure 3.*** In the case you mentioned, EGD indeed significantly accelerates grokking. Blue line (vanilla SGD) achieves accuracy 95% at epoch 1784 (Table 4 in Appendix C), and EGD (orange line) achieves the same accuracy at epoch 115 (Table 4 in Appendix C) which means that it groks 15.5 times faster than vanilla SGD.
>
> ***Results for MNIST.*** We added results of MNIST and an arithmetic task, but with transformers as the encoder, to Appendix D. We did not try IMDB as grokking is not clearly observed in this task in previous works. Grokfast and Omnigrok talk about this task as one in which we can see grokking; however, in the plots, the validation accuracy never reaches high values and it is too noisy. Among the more practical tasks introduced by prior works, we chose MNIST and modulo multiplication for $p=97$ with a transformer as the decoder. For MNIST, please refer to Table 6 and Figures 10 and 11 (EGD outperforms Grokfast). For the arithmetic task, please refer to Table 7 and Figures 12 and 13; the results of EGD are comparable with Grokfast without introducing extra memory usage, which is important as the optimizer tracks momentum and velocity and the architecture is large, 2-layer transformer.
>
>
> ***Code and Reproducibility.*** As suggested by the reviewer, we added Appendix G showing all of the pseudocodes. Refer to Section 5 for experimental setup details of the initial experiments and to Appendix B for the hyper-parameters used for them. For the new experiments, we have discussed the setup and parameters in each appendix separately. We will properly anonymize the code and data gathered for the camera-ready submission.
> We are happy to respond to any further questions the reviewer might have.
>
> We are happy to respond to any further questions the reviewer might have.

---

> > ### Comment · Reviewer_vvi7 · 2025-11-26
> >
> > Thank you for responding to my questions.
> >
> > I appreciate the response to `When does one exactly turn off EGD?`,  that the paper highlights that `RSVD has two important parameters: rank and number of iterations in the inner loop.` and provides empirical evidence that `RSVD` is a suitable alternative. The inclusion of the MNIST and arithmetic tasks in the paper strengthened the findings and improved the comparison with Grokfast. The addition of Pseudo Code and details around the experiments improved the clarity of the paper.
> >
> > I have no further questions.

---

### Official Review · Reviewer_6NV3 · 2025-10-31

**Soundness:** 2
**Presentation:** 3
**Contribution:** 3
**Rating:** 6
**Confidence:** 4

**Summary:**

This article studies the grokking phenomenon based on new insights on simple classification problems. A theoretical analysis is provided on a binary classification problem, showing how grokking can arise due to ill-conditioning in optimization. Then a modified gradient descent method (egalitarian GD) is proposed to adjust the gradient direction to avoid stagnation during training. Numerical results on sparse parity and modular addition also confirm the effectiveness of the insights.

**Strengths:**

-	The theoretical picture of the grokking phenomenon is interesting.
-	The proposed EGD algorithm works well in practice.
-	The article is well written, with a quite complete literature review.

**Weaknesses:**

-  The theoretical example is somehow biased and too low-dimensional, which makes it hard to explain practical results.
- There is no theoretical convergence analysis on the proposed EGD algorithm.

**Questions:**

- Can you explain why you choose a different training and test data distribution in Section 3 (fig 5)? This is not a standard machine learning setup.
- Are you considering the same training and test distributions in the numerical results in Section 5? If so, I would suggest to modify the theory in Section 3 to make it consistent with the setup in Section 5.
- It is unclear how the G is F in  eq. 11 is estimated from mini-batch samples. This practical aspect should be discussed to better understand the numerical results.

---

> ### Author Response · Authors · 2025-11-19
> **Response to reviewer 6NV3**
>
> We thank the reviewer for their great insights. We shall address the main questions and remarks of the reviewer in detail.
>
> ***About our toy tilted Gaussian setup.*** The theoretical example developed in Section 3 is not meant to represent real world models but to capture what we think are important symptoms (1) ill-conditioned gradients along optimization paths, and (2) a nothing to all event. The proposed toy model leads to an exactly solvable setting which can pin-point the grokking behavior of GD, and as a starting point for our optimizer (EGD) proposed in the sequel. In passing this example, produces yet another novel way delayed generalization and eventual grokking can be induced: covariate ill-conditioned + covariate shift.
>
> ***Difference Train and Test Distribution (Toy Setting).*** The rationale behind the construction of this toy example is the following.
>
> - (1) The ill-conditioned covariance structure slows down vanilla GD so that it takes an arbitrarily long time to converge to the OLS (ordinary least squares) solution.This ill-conditioning which greatly affects the behavior of gradients along the optimization path, is controlled by the $\varepsilon$ parameter.
> - (2) The disparity between train and test distribution induces an effect whereby even solutions arbitrarily close to the OLS solution might perform poorly on the test data. This covariate shift is controlled by the $s$ parameter or equivalently, the angle $\theta$ (responsible for the tilting). Once the OLS solution is recovered, the performance switches from mediocre to perfect (a zero-to-all event, put roughly).
>
> This is the easiest way we found to explicitly induce arbitrarily long plateaus, while keeping the setup fully solvable analytically.
>
> We are not claiming that covariate shift is necessary for delayed generalization and then grokking, in general. In fact (1) and (2) above can be induced in different ways, as in arithmetic tasks, for example. The particular way these are induced is not important to the functioning of our algorithm.
>
> ***Test Distributions in Section 5.*** No, in all those experiments there is no covariate shift as in the toy example introduced in Section 3, Still the long plateaus in vanilla (S)GD are happening for similar reasons: (1) ill-conditioning of the gradients along the optimization path (for example, refer to Figure 4), (2) Solutions which generalize perfectly on test have specific symmetrics which must be satisfied (e.g., the embeddings of all data points must organize themselves in a circle, consistent with the Fourier representation of the modular ring $\mathbb Z_p := \mathbb Z/\mathbb Z$ (see Gromov 2023 “Grokking Modular Addition”, and Liu et al. 2022 “Clock and Pizza”). Only at this point does the test performance switch from mediocre to near perfect (zero-to-all event).
>
> ***Link between F and G in Eqn 11.***  Actually, that equation implicitly contains the definition $F=GG^\top$. This has been made more explicit in the revised manuscript. G is a matrix which represents gradient average loss on a **mini-batch**, w.r.t model parameters.
>
> ***Why our toy Gaussian setup focuses on a low-dimensional regime $d=2$.*** Designing our toy Gaussian example in low-dimensions ($d/n \to 0$, concretely $d=2$ and large $n$) allows us to bypass random matrix error analysis which would only lead to technical distractions. It is possible to induce an ill-conditioned design matrix by considering the asymptotic scaling regime $d,n \to \infty$ with $d/n \to \phi$, and the taking $\phi \to 1$. This would qualitatively lead to the same effect: in this regime the condition number of the empirical covariance matrix scales as $\sqrt{n}/(1/\sqrt n)=n = \omega(1)$ (e.g, see Rudelson and Vershynin, “The smallest singular value of a random rectangular matrix”, 2008), which is divergent. All the order core aspects of the analysis would carry over. For this reason, we decided to keep things simple and consider $d=2$, with a population covariance with condition number $1/\varepsilon$, where $\varepsilon>0$ can be taken to be as small as one likes.
>
> ***Convergence analysis of EGD.*** This is highly non-trivial, and is left for future work. We have added this explicitly to the limitations at the end of the Conclusion section. At the moment, we have some extremely promising preliminary ideas for a detailed convergence proof based on ideas around the so-called BPP phase-transition in the theory of spiked random matrix models (e.g., Baik et al., “Phase transition of the largest eigenvalue for nonnull complex sample covariance matrices”, Annals of Probability, 2024).
>
> We are happy to respond to any further questions the reviewer might have.

---

> > ### Comment · Reviewer_6NV3 · 2025-11-25
> > **clarification**
> >
> > Dear authors,
> > I am still confused by your answer, could you clarify this point?
> > * Test Distributions in Section 5. No, in all those experiments there is no covariate shift as in the toy example introduced in Section 3
> >
> > Based on your description on data distribution in Section 3, it seems to me that the training data and test data distribution are not the same, even though they share the same optimal classifier. In Section 5, your answer is that the training and test data distribution are assumed to be the same (no covariance shift), isn't it?
> >
> > My point is to make the theory in Section 3 to make it consistent with the setup in Section 5.
> >
> > best,
> > reviewer

---

> ### Author Response · Authors · 2025-11-28
> **Update: Extension of toy setting to same train and test distribution**
>
> We thank the reviewer for their comment.
>
> We have extended our toy setup so that the train and test distribution are the same, consistently with Section 5 as desired by the reviewer. See **Remark 1** and the second paragraph (in blue) just after **Corollary 1**, in the updated manuscript. The analysis of description and analysis of this equi-distributional setting is provided in **Appendix A.3**.
>
> ### A new theorem and empirical validation
> As with **Corollary 1**, our new **Theorem 2** predicts a plateau in the test error for vanilla GD whose length is of order 1/(eta * epsilon).
>
> In the **Figure 21** illustrates this toy distribution while **Figure 22** illustrates the plateau in vanilla GD predicted by  **Theorem 2**) which is completely removed by our proposed EGD method.
>
> We are happy to address any other questions the reviewer might have.

---

### Official Review · Reviewer_QrdE · 2025-11-01

**Soundness:** 3
**Presentation:** 3
**Contribution:** 3
**Rating:** 6
**Confidence:** 5

**Summary:**

The paper proposes a buffer-free gradient preconditioning method for accelerating grokking phenomenon, reducing the gap between memorization and generalization from spectral analysis. The authors have provided theoretical results that justify their egalitarian gradient descent (EGD) algorithm assuming two-parameter model. Based on the insights, the algorithm is tested on the typical examples where grokking appears, demonstrating the acceleration.

**Strengths:**

- The idea is simple and elegantly addressed, with easy theory and clear empirical examples.
- Experiments show that at least among the given examples, the proposed method effectively reduces the grokking gap.
- The experimental details are sufficient and easily reproducible.

**Weaknesses:**

Some points prevent me from giving a higher score. I will raise my score based on the discussion.

- Since the effect of grokking has demonstrated in various other settings than modular arithmetic such as in [Omnigrok](https://arxiv.org/abs/2210.01117) (MNIST, Transformer, Graph CNN, LSTM) or [Kumar et al. (2024)](https://openreview.net/forum?id=vt5mnLVIVo) (MNIST), more experiments can be performed to further justify the clear effectiveness of the proposed EGD.
- Theoretical justification only exists for toy examples. I believe it should be easy to extend the theoretical results to show that the insights are extendable to larger, more general models such as linear layers or two-layer MLPs.
- The proposed algorithm seems to be similar to RMSProp. Further discussion between the existing gradient descent variants to the proposed method should be carried on. Moreover, how does the proposed egalitarian gradient descent interacts with “momentum” variable which typically appear in the machine learning context?
- The connection between FIM, NGD, and the proposed EGD can be further strengthened by showing how do individual FIM-based and NGD-based algorithms behave under grokking environment. This will further justify the proposed method for accelerating grokking, and imply richer connection between FIM and NGD involved in this problem.

**Questions:**

- The box in line 350-355 seems to swallow additional lines after the equation.
- Some works such as [Kumar et al. (2024)](https://openreview.net/forum?id=vt5mnLVIVo) and [Grokfast](https://arxiv.org/abs/2405.20233) point out that accelerated grokking-induced solution space can be different from the non-accelerated grokking-induced solution space. In other words, the final reaching ground in the parameter space can be differ greatly. Does this proposed method results in the same solution manifold or in the greatly different solution manifold?

---

> ### Author Response · Authors · 2025-11-19
> **Response to Reviewer QrdE**
>
> We thank the reviewer for their great insights. We shall address the main remarks and questions of the reviewer in detail.
>
> ***Additional Experimental Results.*** As the reviewer correctly points out, the grokking phenomenon has been observed beyond mathematical tasks like modular arithmetic and sparse parity problems. Therefore in Appendix D of the updated manuscript, we have added detailed experiments on MNIST dataset (also see Appendix E for experiments on CIFAR-10). The appendix also features experiments on arithmetic tasks, but with transformers (initially we only had MLPs). In all experiments, our proposed EGD method (especially an improved version relying on randomized SVD; see Appendix C) leads to faster grokking times, and beats baselines like Grokfast (which requires much larger memory).
>
> ***Geometry of Solutions.*** Indeed,  accelerated grokking-induced solution space can be different from the non-accelerated grokking-induced solution space. To study the impact of our proposed optimizer, we added experiments in appendix F. In modulo addition and multiplication, the final sub-spaces are different. In sparse parity problems, vanilla SGD tries to reach the same region as EGD, after initially diverging from it (Figures 15–20).
>
> Here is a possible explanation for these observations. For arithmetic tasks like modular addition and multiplication, the space of solutions has a large set of symmetries (due to the periodicity of the optimal solution, which are products of cosine waves with different frequencies; See (Gromov, 2023) and (Doshi et al., 2024). It is conceivable that the different optimizers, having different inductive priors, drive the optimization trajectory to different solutions with the same generalization profile, at the end of training. In contrast, for sparse parity, the solution space has far less symmetries. This drastically constraints the set of generalizing solutions to which an optimizer eventually converges.
>
> Inspired by the above observations, a rigorous study of the geometry of the solutions is left for future work.
>
> ***Comparison with RMSprop.*** In Appendix E, for the CIFAR-10 and non-stationarity experiments, we added EGD to Adam, RAdam, and RMSprop and showed that it improves their ability to handle non-stationarity. Therefore, EGD does not interfere with the optimizers tracking velocity and momentum, and in fact it makes them more agile. The connection between RMSprop and EGD is interesting to be studied in more detail. RMSprop can be seen as a simplified version of EGD as it only normalizes the norms, but EGD tries to normalize all singular values. RMSprop has more similarity with the other simplified version of EGD mentioned in the paper as “Column Normalization”. It would be interesting to study the exact relationship between EGD, Column Normalization, and optimizers tracking momentum and velocity. We leave it to future work.
>
> Also to show more evidence that EGD does not interfere with momentum terms in the optimizers, we want to point out that in Appendix D the optimizers used are Adam and AdamW and both of them have the momentum term. We see that EGD and EGD with randomized-SVD accelerate grokking in those cases as well.
>
> ***Theoretical Analysis in the Case of More General (Non-Linear) Settings.*** The reviewer raises an excellent point. Such an analysis is very difficult and will be communicated in a separate publication once completed. At the moment, we have some preliminary but very promising theoretical results which show how delayed generalization manifests dynamically in the spectral density of the gradient update, and how this connects with ideas from spiked random matrix models. This is already hinted to in Figure 4 of the manuscript.
>
> ***Further Connection to FIM/NDG.*** Conceptually, we don’t expect there to be any deeper connection between our proposed EGD and NDG (natural gradient descent) to be made, beyond what we mentioned in 4.1. Indeed, the design of NDG simply has nothing (no strategic spectral adjusting, etc.) which would catalyze grokking in principle. For this reason, we decided to focus additional experiments on more realistic datasets (MNIST, CIFAR-10), models (linear models, MLPs, transformers), and compare against strong baselines (Grokfast).
> We are happy to respond to any further questions the reviewer might have.

---

### Author Response · Authors · 2025-11-19
**General Response (to all reviewers)**

We would like to thank all reviewers for their insightful comments and suggestions. To supplement the discussion, we have added a vast array of experiments in the appendix, and made other changes (highlighted in blue) to address the points raised by the reviewers. A summary of the main changes is as follows:

- In the initial submission, two figures had limited data in the beginning due to large evaluation intervals. We changed them to show the details better. We also moved the legends of all plots to the top of the figures to open space for the figures.
- We added **Appendix C**, in which we analyze the effect of using RSVD (randomized SVD, a very efficient low-rank approximation) on the arithmetic tasks we studied. In this appendix section, we compare the number of epochs and the wall-clock time to reach high accuracy, and the learning curves. It turns out that if we tune the rank of RSVD properly, our proposed EGD with RSVD is faster than Vanilla SGD, EGD, and the SVD-free simplification of EGD named column normalization in our manuscript.
- We added **Appendix D**, in which we compare between (i) our proposed EGD with exact SVD,  (ii) EGD with RSVD, and (iii) Grokfast, in more practical scenarios. EGD with RSVD outperforms Grokfast on MNIST, in terms of grokking at much earlier epoch and being faster in the sense of wall-clock time. In the studied arithmetic task with transformer model, its performance is comparable while not requiring any extra memory usage.
- To show how EGD can be useful in practical scenarios, we added **Appendix E**, demonstrating that adding EGD to widely used optimizers helps them handle non-stationarity better.
- In **Appendix F**, we added experimental results which analyze whether EGD converges to the same subspace as Vanilla SGD, and if the final solution of these methods are in the same distance from the initialization. For Modulo addition and multiplication, these two converge to different subspaces, but for sparse parity the subspaces are similar. About distance to the initialization, both methods converge to regions with the same radius from initialization.
- We added **Appendix G**, which provides implementation details and pseudo-codes.
- **Appendix A.3** contains an in-depth analysis which extends the toy setup considered in **Section 3** to the case where train and test distribution are equal to the same conically truncated 2D Gaussian. This renders the toy setup completely compatible with real world experiments considered in **Section 5**, as required by Reviewer 6NV3.
- We corrected some typos and small mistakes.

**N.B.:** The efficiency of replacing exact SVD with RSVD in our EGD method was hinted at the end of page 7 of the original manuscript, but we didn’t include empirical justification. This is now provided in the updated manuscript, as explained above.

Below, we address specific points raised by the different reviewers.

---

> ### Author Response · Authors · 2025-12-02
> **Overview of Rebuttal Discussions and Responses**
>
> We would like to thank the area chairs and the ICLR conference organizers for taking proactive measures and accepting extra work to preserve fairness in the decision process in this unprecedented situation. We also would like to again thank the reviewers whose suggestions improved our paper significantly.
>
> As we were at a crucial point when the discussion procedure was stopped, we thought it might be useful to summarize our efforts and the discussions during the rebuttal period to ease going through the dialogue:
>
> **Our Response and revisions**
>
> After reading the insightful comments from the reviewers, we realized that the majority of concerns and questions were focused on the applicability of our method to more complex tasks and architectures. To address these, we added multiple experiments that we had originally left for future work as Appendices C, D, and E. We also went beyond the reviewers’ requirements in Appendix E by showing that our proposed method can help handle non-stationarity, which is important in applications such as reinforcement learning. To address other concerns and questions, we added Appendices F, G, and A.3. The detailed explanation for each appendix is listed in our previous general response and in our responses to reviewers. All added or modified parts are shown in blue in the revised paper.
>
> **Discussion with Reviewers**
>
> *Reviewer QrdE:*
> We addressed all of the questions and weaknesses mentioned by Reviewer QrdE. Unfortunately, we did not receive any follow-up responses from the reviewer during the discussion period.
>
> *Reviewer 6NV3:*
> We were in an active and constructive discussion with Reviewer 6NV3. A few hours before the official stop of the discussion procedure (on 28 Nov 2025), we addressed the final clarification asked by reviewer in Appendix A.3.
>
> *Reviewer vvi7:*
> We addressed all of the questions and concerns mentioned by Reviewer vvi7. In their response on 26 Nov 2025, they indicated their satisfaction with our revisions and answers.
>
> *Reviewer yxV7:*
> We addressed all of the concerns and questions. We also went beyond the original questions in Appendix E by showing that our method can help commonly used optimizers handle high levels of non-stationarity, which is important in various real-world applications such as reinforcement learning. Unfortunately, we did not receive any follow-up responses from the reviewer during the discussion period.

---

### Meta-Review · Area_Chair_69oY · 2025-12-18

**Summary:**

The reviewers indicate the following issues:
- More experiments are needed to further justify the clear effectiveness of the proposed EGD
- Theoretical justification only exists for toy examples
- The proposed algorithm seems to be similar to RMSProp
- The connection between FIM, NGD, and the proposed EGD should be further strengthened
- There is no theoretical convergence analysis on the proposed EGD algorithm
- The consistency between the theoretical analysis (no covariate shift) and empirical analysis (covariate shift)
- A reviewer suggests adding results of this method on Grokking-Induced MNIST and IMBD dataset
- Scalability and computational cost due to the SVD decomposition
- Lack of ablation and sensitivity analysis
- Failure modes or adverse side effects
- Overstated generality and claims beyond the evidence
- Limited theoretical depth on nonlinear/deep networks

**Reviewer Concerns:**

After reading the discussions, I think the following issues are well addressed:
- More experiments are needed to further justify the clear effectiveness of the proposed EGD (the authors have added detailed experiments on MNIST dataset)
- The proposed algorithm seems to be similar to RMSProp (the authors added experiments to show the comparison of the proposed method and RMSProp)
- The consistency between the theoretical analysis (no covariate shift) and empirical analysis (covariate shift) (the authors added more discussions and more analysis to clarify this issue)
- Lack of ablation and sensitivity analysis (the authors have added  ablation and sensitivity analysis to show how randomized-SVD can be used to reduce the computational cost of EGD)
- Failure modes or adverse side effects (the authors added a comparison with Grokfast to clarify the stability issues)
- A reviewer suggests adding results of this method on Grokking-Induced MNIST and IMBD dataset (the authors have added results of MNIST and an arithmetic task)

The following issues are still remaining:
- The connection between FIM, NGD, and the proposed EGD should be further strengthened (the authors do not expect there is any deeper connection between our proposed EGD and NDG)
- Theoretical justification only exists for toy examples (the authors mentioned that such an analysis is very difficult and will be communicated in a separate publication once completed)
- There is no theoretical convergence analysis on the proposed EGD algorithm (the authors think this is highly non-trivial, and is left for future work)
- Limited theoretical depth on nonlinear/deep networks (the authors mention this is very promising and will be explored further in future work)
- Overstated generality and claims beyond the evidence (the authors have not made responses to these issues)

**Reviewer Scores:**

Reviewer QrdE may not change the score since his/her issues on connection between FIM, NGD, and the proposed EGD, and the limited theoretical justifications are not well addressed

Reviewer 6NV3 may not change the score since the authors did not include theoretical convergence analysis on the proposed EGD algorithm

Reviewer vvi7 may not change the score since the current score is very high (8), and all the comments are well addressed

Reviewer yxV7 may not change the score since his/her comment on theoretical depth on nonlinear/deep networks is not well addressed

---

### Decision · Program_Chairs · 2026-01-26

Accept (Poster)